# Unravelling intractable water conflicts: the entanglement of science and politics in decision-making on large hydraulic infrastructure

Jonatan Godinez-Madrigal[1,2] *, Nora Van Cauwenbergh[1], and Pieter van der Zaag[1,2]

[1] Department of Land and Water Management, IHE Delft, Delft, The Netherlands.
[2] Water Management Department, TU Delft, Delft, The Netherlands.

*Correspondence to*: J. Godinez Madrigal (j.godinezmadrigal@gmail.com)

**Abstract.** The development of large infrastructure to address the water challenges of cities around the world can be a financial and social burden for many cities, because of the hidden costs these works entail and social conflicts they often trigger. When conflicts erupt, science is often expected to play a key role in informing policymakers and social actors to clarify controversies surrounding policy responses to water scarcity. However, managing conflicts is a socio-political process, and often quantitative models are used as an attempt to de-politicize such processes; conveying the idea that optimal solutions can be objectively identified despite the many perspectives and interests at play. This raises the question whether science depoliticizes water conflicts, or whether instead conflicts politicize science-policy processes? We use the Zapotillo dam and water transfer project in Mexico to analyze the role of science-policy processes in water conflicts. The Zapotillo project aims at augmenting urban water supply to Guadalajara and León, two large cities in Western Mexico, but a social and legal conflict has stalled the project until today. To analyze the conflict and how stakeholders make sense of it, we interviewed the most relevant actors and studied the negotiations between different interest groups through participant observation. To examine the role of science-policy processes in the conflict, we mobilized concepts of epistemic uncertainty and ambiguity and analyzed the design and use of water resources models produced by key actors aiming to resolve the conflict. While the use of models is a proven method to construct future scenarios and test different strategies, the parameterization of scenarios and their results are influenced by the knowledge and/or interests of actors behind the model. We found that in the Zapotillo case, scenarios reflected the interests and strategies of actors on one side of the conflict, resulting in increased distrust by the opposing actors. We conclude that the dilemma of achieving urban water security through investing in either large infrastructure (supply augmentation) or alternative strategies (demand-side management), cannot be resolved if some key interested parties have not been involved in the scientific processes framing the problem and solution space.

## 1 Introduction

Urban water systems around the world are experiencing various urgent challenges to address water scarcity, flooding, and bad water quality (Zevenbergen et al., 2008; McDonald et al., 2014). The scope of these challenges is such that individual scientific disciplines and traditional approaches fall short of addressing them in a thorough manner to unequivocally inform policy (Funtowicz & Ravetz, 1994; Larsen et al., 2016; Hoekstra et al., 2018). Any solution to the challenges facing urban water

systems will have manifold uncertainties in projected costs, benefits and risks, and this is especially true when large infrastructures are considered (e.g., see Flyvbjerg, 2009 and Crow-Miller et al., 2017, for a general description of the contentious process of cost-benefits assessments of large infrastructures, and for specific cases, see Berkoff, 2003, for China;

Hommes et al., 2016, for Turkey; Hommes & Boelens, 2017, for Peru; and Molle & Floch, 2008, for Thailand). How the perceived costs, benefits and risks are shared among the stakeholders is one of the causes of water conflicts (Delli Priscoli & Wolf, 2009).

Since these conflicts are politically perilous situations, many policymakers seek specialized scientific knowledge that is perceived as neutral and unbiased to serve as the basis of making difficult decisions over controversial issues (Schneider &

Ingram, 1997). In recent years, political ecology literature has acknowledged that this specialized scientific knowledge can act as a form of stealth advocacy in politically charged socio-environmental problems (e.g. Pielke, 2007; Budds, 2009, and Sanz et al., 2019, for groundwater over-exploitation and allocation; Godinez-Madrigal et al., 2019, for water scarcity and surface water allocation). However, literature related to science-policy processes in contexts of intractable conflict due to large infrastructure development is scarce.

This paper has two objectives, 1) to identify the causes of failure in science-policy processes to solve intractable conflicts and promote well-informed water management solutions; and 2) to explore the multiple influences in the production of water knowledge in a context of conflict, and its political use by actors. We contribute to the literature on science-policy process by analyzing the conflict over the Zapotillo dam and water transfer project, perhaps the most politically charged water conflict in Mexico in recent years. This case is of special relevance due to what is at stake: the water supply for the two most important

cities in Western Mexico, the economic importance of its semi-arid donor basin, and the possible displacement of three communities lying in the reservoir's area. Furthermore, the conflict can be considered intractable, given its length (started more than 15 years ago) and that is still largely unresolved due to the intransigent positions of the stakeholders (Putnam & Wondolleck, 2003). The focus of this paper is the scientific knowledge produced through a water resources model developed by an independent international team of experts convened by UNOPS (United Nations Office for Project Services), hereafter

referred to as the UNOPS team, as a means to clarify controversies, fill gaps in knowledge and depoliticize the Zapotillo conflict. We demonstrate how the process of scientific production, in spite of its intended neutrality, favored the Zapotillo project, ignored alternatives proposed by the dam-affected stakeholders based on demand management strategies in the recipient cities, and improperly managed core uncertainties related to climate change and future water demand.

The paper is structured as follows. The first section analyzes the literature on science-policy processes in relation to epistemic

uncertainties and controversies in water conflicts. We then describe the study area and the methods used to analyze the conflict. Subsequently, in the results section, we first describe the trajectory of the regions that would benefit from the Zapotillo project; we then describe the main knowledge uncertainties and controversies that articulate the positions and frames of the actors in conflict; and subsequently we analyze the scientific products that were developed to support decision-making in the conflict. Finally, we discuss the theoretical contributions of the case to the literature of the role of science-policy processes in water

conflicts.

## 2 Science-policy processes and water conflicts

### 2.1 Uncertainties and ambiguity in science-policy processes

Effective science-policy processes in water management are those where water knowledge informs decision-makers as to what are the most appropriate solutions to water challenges, and what is likely to happen if nothing is done (Karl et al., 2007). However, Funtowicz & Ravetz (1994) have argued that complex socio-environmental issues (e.g., climate change) are confronted by uncertainties, ethical complexities, and policy riddles regarding societal values, from which no clear-cut policies can be concluded.

Uncertainties consist not only on matters of lack of precision and accuracy in the data being analyzed, but also of epistemic uncertainties, related to the functioning of a given system (Funtowicz & Ravetz, 1990; Di Baldassarre et al., 2016; Cabello et al., 2018) and of ambiguity, understood as the "simultaneous presence of multiple valid and, sometimes conflicting ways, of framing a problem." (Brugnach & Ingram, 2012). Scientists cannot address these levels of uncertainty by simply improving their techniques or computational prowess (Di Baldassarre et al., 2016). Epistemic uncertainties and ambiguity are entangled with controversies of what the real problem is and how to frame the solutions in the political arena between actors with different interests (Gray, 2003; Cabello et al., 2018).

When facing epistemic uncertainties in a complex socio-environmental problem, stakeholders stand on unexplored territory; even scientists face an ambiguous path in deciding which methodologies to use and how to interpret the phenomena (i.e. Melsen et al., 2018, and Srinivasan et al., 2018; see also Brugnach & Pahl-Wostl, 2008). Boelens et al. (2019) noted the relation of knowledge and power asymmetry between stakeholders in the context of large infrastructural schemes. Such asymmetry is characterized by hegemonic discourses that privilege technical knowledge as infallible, while other kinds of knowledge are disregarded to understand a socio-environmental problem (Schneider & Ingram, 1997; Wesselink et al., 2013). This may result in what Boelens et al. (2019) denominate 'the manufacture of ignorance', understood as the process of cherry-picking facts and knowledge to further one´s position, while discrediting ex-ante competing knowledge without a thorough debate (see also Flyvbjerg, 2009, Moore et al., 2018). In the case of large infrastructures, governments undertake this process often by invoking scientific evidence (Brugnach et al., 2011), which is often presented a-critically by downplaying the inherent risks and uncertainties (Flyvbjerg, 2009), and by presenting it as the only valid frame to understand socio-environmental problems.

When science-policy debates ignore intrinsic epistemic uncertainties and ambiguity, it is expected that uncertainty be present in their scientific recommendations to policy (Funtowicz & Ravetz, 1994), which makes such recommendations dubious, or at least contestable. Alternatively, Pielke (2007: 17) proposed that the role of scientists in issues of high uncertainties and politicization should be that of "honest broker of policy alternatives", consisting of expanding the scope of alternatives to

decision-makers. Moreover, epistemic uncertainties and ambiguity can be made manageable through bottom-up approaches[1] consisting of the inclusion of local stakeholders, their knowledge, problem-framing and alternative solutions in the policy debates (for a general description see Brugnach et al., 2011, and for hydrological risk management see Lane et al., 2011, and Blöschl et al., 2013). Nevertheless, public participation in socio-environmental decisions is a political decision often aimed at improving the acceptability and legitimization of policies (Newig, 2007), rather than reducing epistemic uncertainty and handling ambiguity (Blomquist & Schlager, 2005; Brugnach & Ingram, 2012). In such situations the underlying causes for conflict remain un-addressed.

## 2.2 Water conflicts and co-production of knowledge

Water conflicts emerge for many reasons, but we will explore those that emerge from the imposition of large infrastructural projects. These projects may produce many benefits, but also socio-environmental costs and risks that are unevenly distributed between stakeholders. An example is the apparent urgency to implement supply augmentation and reallocation solutions to guarantee water supply to large cities. These solutions may hamper due processes of transparency, public participation and the rights of other water users and stakeholders. The absence of these processes may create social conflicts (Barraqué & Zandaryaa, 2011; Roa-García, 2014), which are defined as "two or more entities, one or more of which perceives a goal as being blocked by another entity, and power being exerted to overcome the perceived blockage" (Frey, 1993, cited in Delli Priscoli & Wolf, 2009). Thus, water conflicts may block such supply augmentation projects to alleviate water scarcity, while no alternative solutions are implemented. In doing so, actors in conflict may worsen the system as a whole (Madani, 2010), aggravating the social conditions by rationing water, and deteriorating hydrological conditions by further depleting available water reserves like aquifers or dams.

When these conflicts are prolonged in time, the positions of the actors in conflict tend to harden and the conflict may become intractable with small chances for a negotiated solution (Putnam & Wondolleck, 2003). Intractable conflicts are often characterized also by ambiguity, in which actors with different systems of knowledge (engineers, communities, policymakers, etc.) perceive the problem with different frames, as well as its possible solutions (e.g. Table 1 presents the multiple frames of the actors in the Zapotillo conflict). A diversity of frames is possible since water problems are often unstructured and riddled by uncertainties in information and cause-effect relationships (Islam & Susskind, 2018). Even within stakeholder groups, stakeholders can make sense of the conflict using different frames (Brummans et al., 2008). Politicians typically expect scientists to contribute to unravelling what the problem is, and to offer solutions supported by all actors (Schneider & Ingram, 1997). However, studies have identified political biases in allegedly neutral scientific studies (i.e., Budds, 2009; Milman & Ray, 2011; Fernandez, 2014; Sanz et al., 2018; Godinez-Madrigal et al., 2019), which have lately discredited science as a fair

---

[1] The difference between a top-down and a bottom-up approach is that the first focuses on highly technical assessments, while the second on the communities' vulnerabilities, making the latter more robust to a changing and unpredictable climate, no matter how low the probabilities of the occurrence of any event (Blöschl et al., 2013).

knowledge creator in some controversial large infrastructural water projects around the world (Boelens et al., 2019). Due to this situation, among others, more attention has been given to include stakeholders in research and decision making (Armitage et al., 2015; Krueger et al., 2016).

Specialized literature provides some consistent recommendations regarding knowledge in contexts of conflict and a diversity of values in socio-environmental problems. Van der Zaag & Gupta (2008) recommend to consider five principles based on feasibility, sustainability, considering alternatives, good governance and respecting rights and needs before undertaking large infrastructural schemes; Funtowicz & Ravetz (1994), Van Cauwenbergh (2008), Islam & Susskind (2015), Armitage et al. (2015) Dunn et al. (2017) and Norström et al. (2020) argue that since no expertise or discipline can claim to have the monopoly of wisdom in complex socio-environmental issues, the problem definition and possible solutions need to include local and non-technical knowledges, therefore engaging in co-production of knowledge. This approach even provides the advantage of designing more robust and resilient solutions (Blöschl et al., 2013). This does not belittle scientific studies, but changes their role to become boundary objects, which cannot illuminate stakeholders´ decision-making, but rather elicit new relationships and innovative solutions among the different systems of knowledge and frames present in all stakeholders (Lejano and Ingram, 2009). True knowledge controversies have the potential to be generative events in the sense that they open the ontological question of what is reality and how it is framed, and redefine it in, hopefully, better terms (Callon, 1998; Latour, 2004; Whatmore, 2009).

However, little attention has been paid to science-policy processes in cases of intractable water conflicts based on the development of large infrastructures to solve urgent water problems. The next sections present the historical context of the conflict over the Zapotillo water transfer project in Mexico, analyze the knowledge controversies around the conflict and the scientific products developed by team of experts fielded by UNOPS and by Conagua (the federal water authority) to solve the conflict and generate acceptance and legitimacy for the project.

## 3 Case study and Methods

### 3.1 Study areas

Since the Zapotillo project entails the water transfer from the Verde River Basin in the northeast of Jalisco to two cities located outside of the boundaries of the basin, three different regions constitute the area of interest of this study. Figure 1 shows the two recipient cities of the projected water transfer, Guadalajara and León, and the contiguous donor basin, the Verde River Basin. Currently, Guadalajara has more than 4.5 million people, and is the capital of the State of Jalisco. León has a population of around 1.5 million people and is the most populous and economically most important city of the State of Guanajuato.[2] The Verde River Basin is a sub-basin of the Lerma-Santiago-Pacífico basin and discharges its water to the Santiago River located north-west of Guadalajara. The area of this sub-basin is around 21,000 km$^2$ large and is mainly located in the State of Jalisco

---

[2] For further information on Guadalajara and León, consult supplementary material.

(55%). The sub-basin is considered as semi-arid in the north, with an average precipitation of around 360 mm/year, and sub-
tropical in the south with an average precipitation of 900 mm/year; the average temperature varies between 11ºC and 18ºC in
winter and 17ºC and 25ºC in summer; and the average potential evaporation in the basin is around 1550 mm/year (UNOPS,
2017a). The basin is home to around 2 million people, of which almost half inhabit the region of Los Altos, located in the part
of the basin that belongs to the State of Jalisco. The northern part of the basin, located in the State of Aguascalientes, is
160 characterized by a developed industrial sector; while Los Altos is characterized by a vibrant primary sector of the economy,
contributing to the production of around 20% of the total animal protein produce of the country (Ochoa-García et al., 2014).

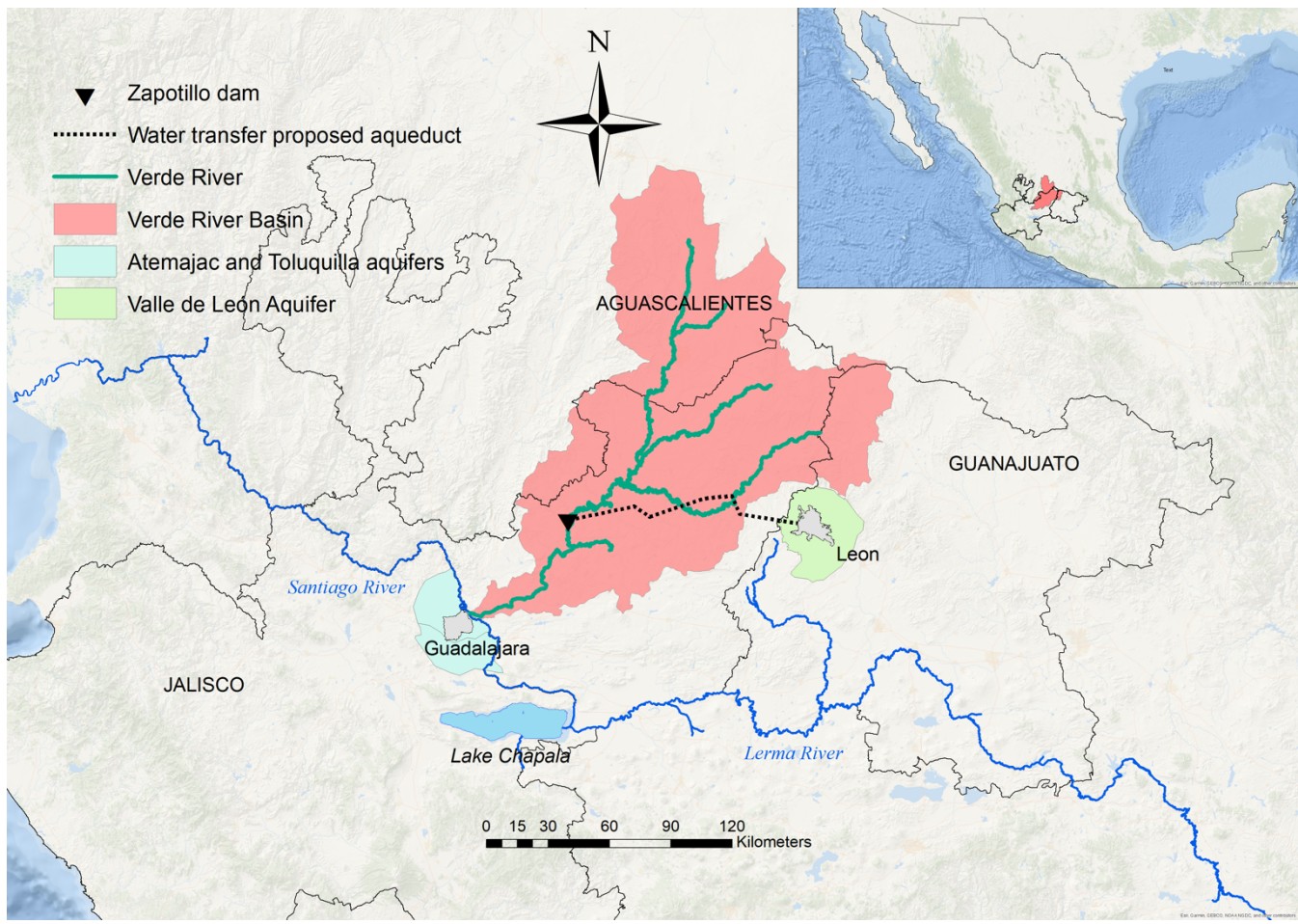

**Figure 1: Map of the Verde River Basin and main cities (Source of GIS layers: © 2018 Conagua, and © 2019 Esri, Garmin, GEBCO, NOAA NGDC, and other contributors).**

**3.2 Methods**

To understand the science-policy processes in a context of an intractable conflict we adopted an interdisciplinary method to comprehensively analyze the technical as well as the social issues that are central to the conflict. The first author spent five months before the public release of the report by the UNOPS team in Guadalajara in 2017 and one month after. He conducted 22 in-depth, semi-structured interviews to most of the key actors of the conflict: members of Jalisco's government, national and state water authorities, NGOs, scholars, the Citizen Water Observatory (hereafter referred to as the Observatory) and representatives of the communities affected by the dam. Since the hotspot of the conflict was located in Jalisco, we decided to focus on Jalisco instead of Guanajuato; although we also collected information on Guanajuato through many actors in Jalisco that had close contact with key stakeholders in Guanajuato and through public statements and official documents of the local water utility and state water authorities. The semi-structured interviews consisted of exploring three main themes: the root causes of the problem and the conflict, what were the sources of controversy in the conflict, and what would be the preferred solutions to the conflict and the water scarcity problem. The interviews also served to identify the position and interests of the actors in the conflict after Fisher et al. (2000) that in turn allowed differentiation of stakeholders following Reed at al. (2009). Due to the delicate nature of the situation, all interviewees remain anonymous, and not all interviews could be recorded; in such cases we relied on fieldnotes taken immediately after the interview. The interviews that were recorded, were transcribed. We analyzed the interview transcripts and fieldnotes to extract the summarized viewpoint of the stakeholders, which are described in Table 1. We then conducted participant observation during five key meetings of the Observatory and Jalisco's government to analyze the discourses, knowledge claims, and main controversies on the coupled human-water system of the region. This allowed us to identify controversies and link the position of actors in the conflict to knowledge frames. Immediately after the presentation of results from the study by UNOPS' team, we conducted informal interviews with most of the key actors that were present, to chronicle in our fieldnotes their reactions and opinions on the outcome of the study.

Afterwards, we requested from Jalisco´s government the full water resources model that the UNOPS team developed; we received it by the end of 2017. The model was developed using the Water Evaluation and Planning System (WEAP21) software (see supplementary material for a detailed description of the model), and contained the five scenarios that the UNOPS team used to test the viability of the Zapotillo dam project to reliably allocate water until the year 2069 (Figure 2). The five scenarios switched parameters under different reservoir storage volumes (at dam heights 80 m and 105 m), different water allocation volumes to Guadalajara, León, and the urban localities within the Verde River Basin (three aggregated flows of water were considered: 8.6 m³/s, 4.8 m³/s and 7.5 m³/s; Figure 2 disaggregate these flows to the three users), changes in water availability related to climate change (RPC 8.5 or no climate change) and changes in agricultural water demand in the donor basin (static water demand since year 2018 or expected water demand in year 2030).

The UNOPS team recommended decision makers that the best possible configuration of the Zapotillo project was that of scenario 5: to build a dam at 105 m, with the only caveat of reducing the water allocation by 13%. However, many actors were negatively surprised that although the UNOPS team developed a scenario with climate change and future water demand (scenario 4, see Figure 2), these changing future conditions were not included in their scenario 5, which only considers current water demand and ignores reduced water availability due to climate change. Therefore, we considered it important to replicate

the results developed by the UNOPS team, and to test and analyze its choice of scenarios and recommendation by developing an additional scenario (our) that included the variables climate change and future water demand as developed by the UNOPS team in scenario 4 to their scenario 5 (Figure 2). We then compared the results of our scenario with the original scenario 5 using the same indicators the UNOPS team used to assess their own scenarios. These indicators (reliability, vulnerability, and resilience) were based on the methodology of Loucks and Gladwell (1999). Reliability assessed the percentage of months the dam was able to supply its intended volume. The ideal score would be 100%. Vulnerability assessed the percentage of water supplied vis-à-vis water demand for all months. The ideal score would also be 100%. And resilience assessed the speed of recovery of the dam after a period of being empty by calculating the number of times a satisfactory value (when all water demand is satisfied) follows an unsatisfactory value (when not all water demand is satisfied) divided by the number of unsatisfactory values. The scores range from 1 to 0, being close to 1 represents a highly resilient system, and 0 a poorly resilient system.[3]

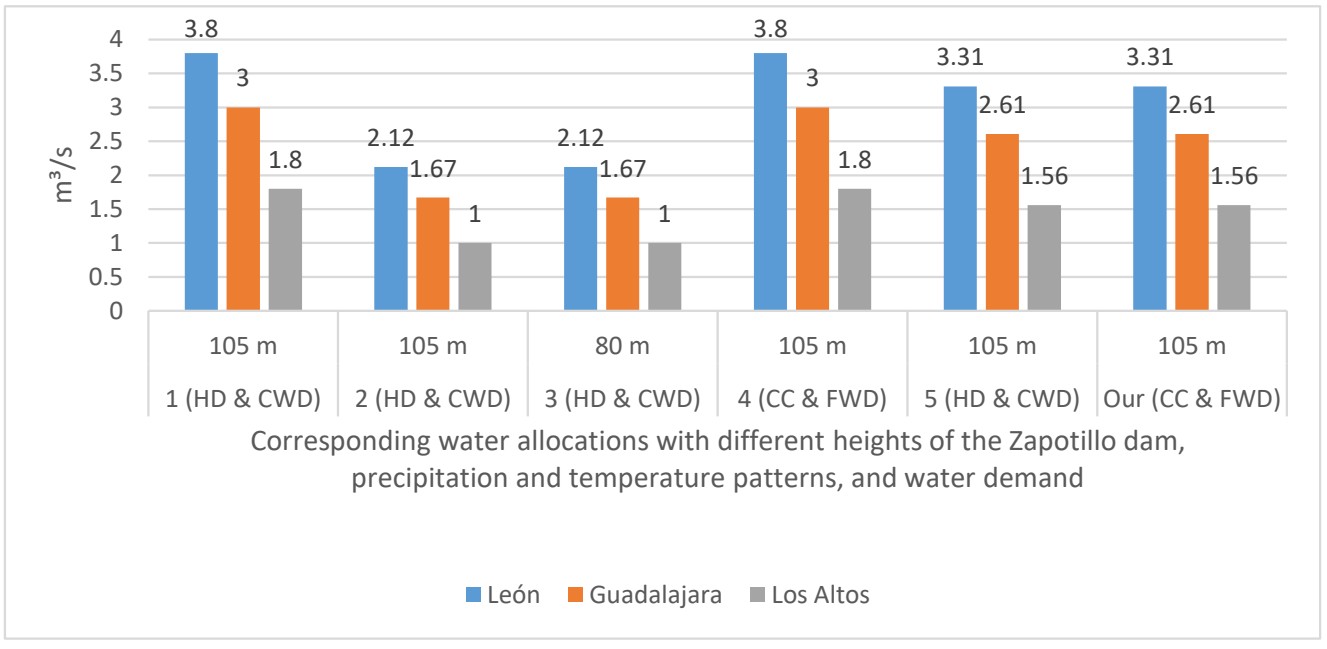

**Figure 2: Key variables of the five water allocation scenarios (in m³/s for León, Guadalajara and Los Altos) developed by UNOPS (2017b) and ours ("HD & CWD" = historical run-off data and current water demand; "CC & FWD" run-off under climate change and future water demand).**

## 4. Results

### 4.1 The Zapotillo conflict

---

[3] The resilience indicator is only useful when the system presents unsatisfactory values, therefore if the system does not present any unsatisfactory values, the indicator is non-existent, as seen in Figure 6 (below).

Guadalajara and León are the most important cities of their respective States, Jalisco, and Guanajuato, in terms of population and economic size. Since the 1950s, Guadalajara's local water resources availability was overrun by the increasing water demand, and water managers sought to increase its water supply from Lake Chapala, the largest lake in the country. Currently,

Guadalajara complements its water demand mainly through groundwater (see Table S1 in the supplementary material). However, due to their intense use, the aquifers are considered as over-exploited and with presence of nitrate and sulphate due to farming activities and wastewater disposal, and naturally occurring contaminants like lithium, manganese, fluorine, and barium due to mixing of hydrothermal fluids (Hernandez-Antonio et al., 2015; Mahlknecht et al., 2017; Moran-Ramirez., 2016). León, on the other hand, does not have large bodies of surface water in close vicinity and therefore it has historically

relied solely on groundwater, which is now considered as heavily over-exploited with a drawdown of 1.5 m/year and with presence of chromium due to industrial activities, related to anthropogenic activities nitrate, chloride, sulphate, vanadium and pathogens, and naturally occurring contaminants like fluoride, arsenic, iron, and manganese due to the introduction of older groundwater with longer residence times (Esteller et al., 2012; Villalobos-Aragon et al., 2012; Cortes et al,. 2015; SAPAL, 2020).

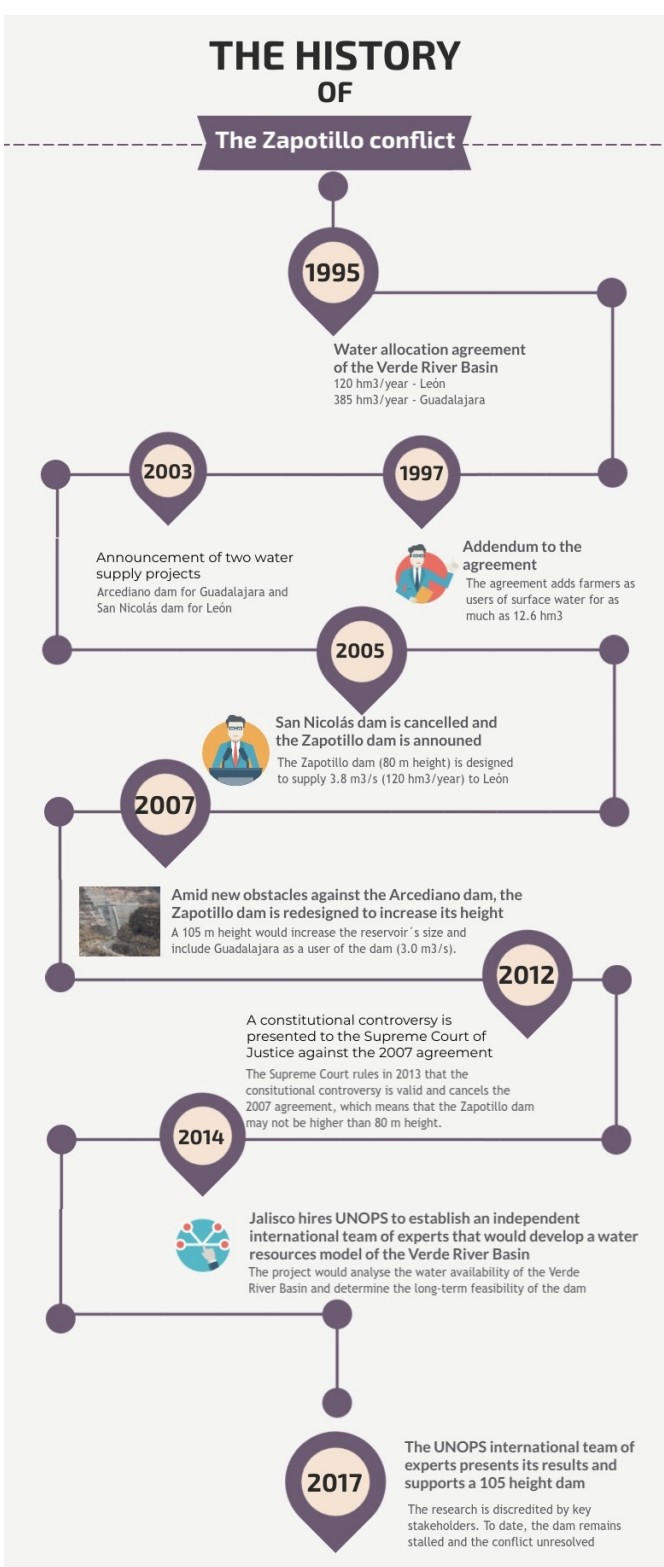

## THE HISTORY
### OF
**The Zapotillo conflict**

**1995**

Water allocation agreement
of the Verde River Basin
120 hm3/year · León
385 hm3/year · Guadalajara

**2003**

Announcement of two water
supply projects
Arcediano dam for Guadalajara and
San Nicolás dam for León

**1997**

Addendum to the
agreement
The agreement adds farmers as
users of surface water for as
much as 12.6 hm3

**2005**

San Nicolás dam is cancelled and
the Zapotillo dam is announed
The Zapotillo dam (80 m height) is designed
to supply 3.8 m3/s (120 hm3/year) to León

**2007**

Amid new obstacles against the Arcediano dam, the
Zapotillo dam is redesigned to increase its height
A 105 m height would increase the reservoir´s size and
include Guadalajara as a user of the dam (3.0 m3/s).

**2012**

A constitutional controversy is
presented to the Supreme Court of
Justice against the 2007 agreement
The Supreme Court rules in 2013 that the
consitutional controversy is valid and cancels the
2007 agreement, which means that the Zapotillo dam
may not be higher than 80 m height.

**2014**

Jalisco hires UNOPS to establish an independent
international team of experts that would develop a water
resources model of the Verde River Basin
The project would analyse the water availability of the Verde
River Basin and determine the long-term feasibility of the dam

**2017**

The UNOPS international team of
experts presents its results and
supports a 105 height dam
The research is discredited by key
stakeholders. To date, the dam remains
stalled and the conflict unresolved

**Figure 3. Timeline of the Zapotillo conflict.**

During the 1980s, water managers in Jalisco were aware of the relentless growth of Guadalajara and sought to develop new sources of water besides groundwater and Lake Chapala (Flores Berrones, 1987). They analyzed that the only nearby region with enough water to supply Guadalajara was the Verde River Basin, located in the north of Jalisco (Figure 1). They calculated a potential of more than 20 m³/s, enough to supply water for Guadalajara for the coming decades. However, it was technically complicated to develop the Verde River Basin and transfer its water to Guadalajara. The Verde River discharges into the Santiago River at around 500 meters below the altitude of Guadalajara, which skyrockets pumping energy costs. During the 1990s Jalisco developed many projects that failed to materialize due to financial and political issues (Von Bertrab, 2003). During this time and partially because of the inability of Jalisco to materialize a water transfer project, Guanajuato requested Conagua (the federal water authority) legal rights over a portion of the Verde River's water for the city of León. In 1995, Conagua accepted this request and added Guanajuato as a potential user of the river's water.

During the year 2000, a drought started in the Lerma-Chapala basin that caused a water crisis for Lake Chapala, which decreased its volume to less than 10% of its capacity. Since Guadalajara heavily relied on the lake for its water supply and upstream farmers in Guanajuato used most of the surface water that fed the lake, the situation triggered a surface water allocation conflict between Jalisco and Guanajuato (Godinez-Madrigal et al., 2019). The conflict was resolved by reducing the water rights of upstream farmers to increase the volume of water reaching the lake. But, in exchange, in 2003 Conagua promised to build the San Nicolás dam in the Verde River Basin to transfer water to León, and the Arcediano dam in the Santiago River for Guadalajara (Godinez-Madrigal et al., 2019).

After a swift mobilization of the San Nicolás community, the dam was cancelled in 2004. However, in 2005, the Zapotillo project was unveiled, it was designed at 80 m height with the objective to provide 3.8 m³/s only to León. It is at this moment in time when the authors pinpoint the start of the Zapotillo conflict, which is summarized in Figure 3. Nevertheless, because the water authorities could not solve important social, financial and technical issues to build the Arcediano dam (López-Ramírez & Ochoa-García, 2012), Jalisco´s government advocated in 2007 to change the design of the Zapotillo project to include Guadalajara as a user and receive 3.0 m³/s by increasing the dam's height to 105 m to increase its storage capacity.[4]

By this time, the dam-affected communities, Temacapulín, Acasico and Palmarejo (hereafter Temacapulín), had already started a fierce opposition against the project with the objective to avoid the flooding and relocation of their communities. Their representatives followed a social and legal strategy, which consisted of claiming that the 2007 agreement was unconstitutional because Jalisco's governor did not consult the State congress. In 2013, the Mexican Supreme Court ruled against the 2007 agreement and ordered Conagua to stop the construction of the dam, which by then already had reached 80 m height (DOF, 2013). The Zapotillo project has remained paralyzed since then. Although the dam wall has already been built, the reservoir has not been filled, because of the uncertainty of the dam's final height.

---

[4] Several urban locations in the Los Altos region were included as well in the water allocation agreement of the project, which would receive 1.8 m³/s.

Given the politicization of the conflict and the urgency of meeting the water deficits of Guadalajara and León without implementing any additional or alternative strategy, new actors have entered the political arena (see Figure 4 for a comprehensive map of actors of the Zapotillo conflict). Some farmers' associations of Los Altos coalesced and lobbied against
the Zapotillo project using the argument that the region is semi-arid, already presents groundwater over-exploitation, that climate change will worsen the condition of the regional water resources, and that the region is one of the most productive agricultural regions in the country (Ochoa-Garcia et al., 2014). Additionally, due to the increased political pressure, in 2014 Jalisco's government supported the creation of a Citizen's Water Observatory, led by an active spokesperson of farmers of Los Altos, and composed of a wide range of representatives of universities and civil society organizations (see supplementary
material for more information) that would, at least in theory, have the mandate to formulate binding recommendations to local and state governments of Jalisco. The Observatory, NGOs and local universities argued that demand management strategies in Guadalajara and León could be more sustainable and socially just than the Zapotillo project. In contrast, IMTA (the engineering body of Conagua) released a technical study concluding that the Zapotillo project was feasible (there was enough water availability in the basin) even in the context of climate change (IMTA, 2015).

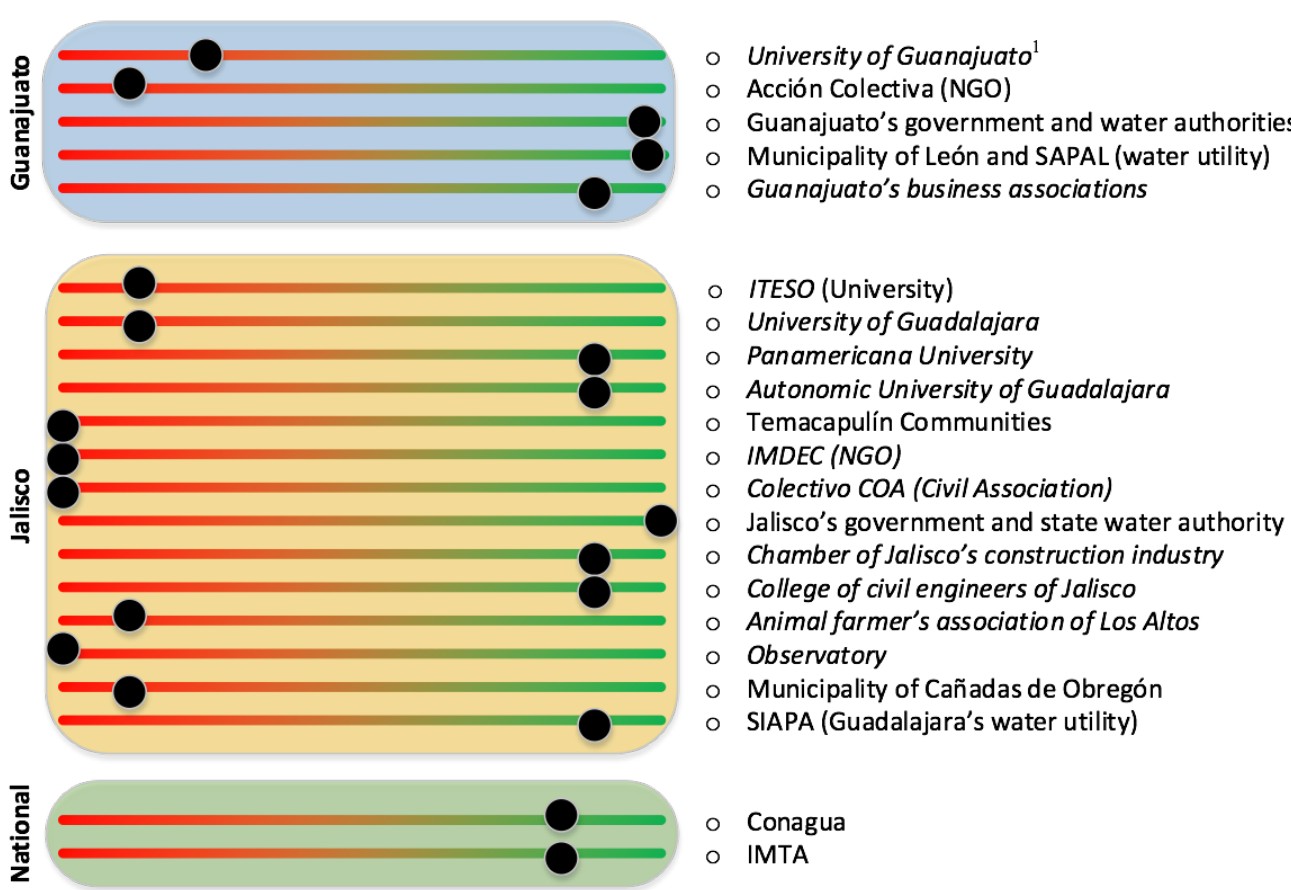

- University of Guanajuato[1]
- Acción Colectiva (NGO)
- Guanajuato's government and water authorities
- Municipality of León and SAPAL (water utility)
- Guanajuato's business associations

- ITESO (University)
- University of Guadalajara
- Panamericana University
- Autonomic University of Guadalajara
- Temacapulín Communities
- IMDEC (NGO)
- Colectivo COA (Civil Association)
- Jalisco's government and state water authority
- Chamber of Jalisco's construction industry
- College of civil engineers of Jalisco
- Animal farmer's association of Los Altos
- Observatory
- Municipality of Cañadas de Obregón
- SIAPA (Guadalajara's water utility)

- Conagua
- IMTA

[1]Universidad de Guanajuato has not released any official position on the project, however many of its academics have publicly supported its cancelation.

**Figure 4. Position of key actors on a horizontal axis against (left, red) and in favor (right, green) the Zapotillo dam project, and new actors are highlighted in italics (for more details on the Figure methodology and description of actors see Table 2 in the supplementary material).**

In 2014 Jalisco's government hired the United Nations Office for Project Services (UNOPS) to establish an independent international team of experts tasked to develop a water resources model of the Verde River Basin and formulate an informed recommendation to address, once and for all, the controversies regarding the possible negative effects in the Verde River Basin and analyze the optimal configuration of the Zapotillo project. The involvement of UNOPS was immediately seen as an existential threat to the recently created Observatory, because the latter assumed as its primary function to determine the future of the Zapotillo project and recommend actions to solve the conflict. In fact, the chair of the Observatory criticized the involvement of UNOPS as a political play by Jalisco´s government to decrease the Observatory´s influence. He also questioned the integrity of the UNOPS´ team due to the apparently suspicious high cost of the study (4.5 million USD); and refuted *ex-ante* the technical study of the UNOPS´ team. Based on these criticisms, the leadership of the Observatory lamented that Jalisco´s government had not funded them and the University of Guadalajara instead to do the research. However, a high-level official of Jalisco´s government (personal comm. 22/05/2017) characterized the criticisms from the Observatory as representing the political interests of the University of Guadalajara, who often lobby Jalisco´s government to receive more financial resources (Jalisco´s government determines the University´s budget) and research contracts. Moreover, Jalisco´s government had previously awarded environmental research projects to academics of the Universidad de Guadalajara, but, according to the official, the resulting studies were technically deficient and unusable. Related to IMTA, the appreciation of this official is that its function has been relegated to technically legitimize Conagua´s projects, and that it was reluctant to share any information. The official concluded that "the scientific debate is very poor, because it has been co-opted by politics." This explains why Jalisco's government neither trusted the University of Guadalajara nor IMTA and that it approached UNOPS as an alleged apolitical third party with proven independence (UN-affiliated) and technical capabilities that were locally absent to help solve the conflict. The government official said that "[Hiring] UNOPS will articulate a paradigmatic change in the way we make decisions on water management in Jalisco."

The UNOPS´ study took two years, and the process followed and methods adopted were largely unknown by most actors. Finally, in 2017, the UNOPS team of experts recommended that the Zapotillo dam should be built at 105 m height and that the original water allocation should decrease by 13%, since Conagua had over-estimated the available water in the Verde River Basin and underestimated water demand (UNOPS, 2017c). The results of the study were discredited and discarded by some of the main stakeholders in the conflict as described in Section 4.3.

**4.2     Controversies**

Table 1 summarizes the main controversies and frames raised by the interviewed actors in the conflict. These can be divided into two: 1) what are the appropriate policies to solve the water scarcity problems in the recipient basins (Guadalajara and León); and 2) what are the risks, uncertainties and negative effects of a dam and a water transfer in the Verde River Basin, the donor basin.

**Table 1. Main controversies and frames on the coupled human-water system of the regions and the Zapotillo project (ZP).**

| General controversies | Specific controversies | Frames |
|---|---|---|
| **Recipient basins: policies for urban water security** | – The urgency to apply supply augmentation policies to achieve water security. <br> – Replacing supply-side policies for demand management policies and small-scale infrastructure: reducing physical losses in the network and implementing rainwater harvesting. <br> – Financial burden because of increasing unexpected costs of large infrastructure. <br> – Alternative, in-basin water sources for León and Guadalajara. <br> – Sectoral water transfers to reduce groundwater over-exploitation. | – Actors in favor of ZP: alternatives are unrealistic. ZP is the only feasible solution to achieve water security. <br> – Actors against ZP: Alternatives exist and can be cheaper, more sustainable, and socially just than ZP. |
| **Negative consequences for the donor basin** | – Dam's height in relation to the resettlement of the three communities and the water allocation commitments to León and Guadalajara. <br> – Overestimation of surface run-off in the Verde River Basin. <br> – Future water scarcity due to droughts and climate change in the Verde River Basin. <br> – Underestimated official water abstractions in the Verde River Basin. <br> – Regional socio-economic dynamic is growing, as well as water demand in the Verde River Basin. <br> – Current groundwater over-exploitation will increase in the future. <br> – The human rights of Temacapulín should be respected. | – Actors in favor of ZP: There is enough water in the donor basin for all existing and future users. And a 105 m height dam is the best and most efficient solution that benefits a great majority despite the social costs of relocating Temacapulín. <br> – Only a 60 m height dam is socially feasible, since human rights are not negotiable. <br> – Actors against ZP: There is currently not enough water in the donor basin, and a water transfer will have enduring negative effects for the region. |

**4.2.1 Recipient basins: policies for urban water security**

Since the 1980s, Guadalajara's per capita water use has remained above 200 l/cap/day (Flores-Berrones, 1987; Consejo Consultivo del Agua, 2010). Ever since, water authorities have strived to keep pace with the fast-growing city population, because they consider a relatively large per capita water use as an important indicator for water security. In a context of a decreasing per capita water availability because of population growth, the actors in favor of the Zapotillo dam project have emphasized the urgent necessity of supply augmentation for the cities of León and Guadalajara. Representatives from CEA-

Jalisco (Jalisco's water authority) and Sapal (León's water utility) argued that without supply augmentation, both cities might suffer a water crisis due to water scarcity derived from the over-exploitation of its aquifers. Water authorities from Jalisco and Guanajuato concluded that pressure on aquifers in both cities and Lake Chapala need to be decreased, as aquifers represent a safe backup in times of drought. An additional risk for Guadalajara is the aging Lake Chapala aqueduct, whose life expectancy has already been exceeded. Repairing the aqueduct may affect the water supply for the city for weeks or even months.

Against this argument, representatives of Temacapulín, the Observatory, NGOs and universities have argued that supply augmentation will always lag behind water demand. This group of opposing actors argues that there is an urgent need to curb the per capita water use, and to limit the cities' physical expansion and demographic growth, supported by a transition to a demand management paradigm that considers a reduction of physical losses, development of alternative water sources like rainwater harvesting, sectoral water transfers and full cost recovery of water utilities.

Regarding urban rainwater harvesting, a group within the Universidad de Guadalajara (not a member of the Observatory) has been developing and promoting this solution over the last decade (Gleason-Espíndola et al., 2018). They claim that harvesting rain through household systems distributed across the city could eventually make unnecessary a supply-augmentation project such as the Zapotillo project. However, according to their own estimates, the proposed system could harvest approximately 21 hm³/year, which could account for only about 7% of the total water use of Guadalajara, which is 313 hm³/year (SIAPA, 2017).

Researchers at the University of Guanajuato calculated an approximate annual rainwater harvest of 27.3 hm³/year for the city of León, amounting to 33% of the total water use of 81 hm³/year (Tagle-Zamora et al., 2018). It should be noted, however, that both studies differed in their methodology and approach, and both did not account for implementation uncertainties, a reason for Jalisco´s water authority to dismiss rainwater harvesting as a realistic option.

The Observatory has argued that the municipality of León and the government of Guanajuato should integrate their water
resources at the basin scale to save water and reallocate it to where it is most needed. For this, Jalisco´s Observatory proposed a two-way strategy for León: to abstract water from Sierra de Lobos, a mountain range located close to León, and to implement an agricultural water modernization program and to reallocate its savings to León. The Observatory claims such a strategy would increase available water for León with 360 hm³/year, which is four times León's current water use (Del Castillo, 2018). However, even after request, the technical details of this alternative have not been shared nor made public anywhere. In fact,

a member of the Observatory recognized that the technical members of the Observatory produce these claims based on "feeling" rather than on technical analysis (personal comm. 08/05/2017).

When looking at a reduction of physical losses, Fitch Ratings (2015) stated that the current losses of Guadalajara's distribution system account for more than 3 m³/s (around 32% of distributed flow). Gómez-Jauregui-Abdo (2015) warned that this situation

may worsen, because of the network's obsolescence rate, which is higher than the replacement rate. CEA-Jalisco has argued
that Siapa's budget is not sufficient to replace the entire distribution system and that even if sufficient financial resources were
available it would imply a huge social cost by breaking the asphalt of the streets of the whole city and paralyze the traffic. This
would also imply a political cost that no local politician is willing to assume. In León, Sapal's non-revenue water also amounts
to approximately 32%. Although the replacement rate of their distribution system is higher than Guadalajara's, their
distribution system's deterioration rate is not precisely known.

Representatives of CEA-Jalisco consider all these alternative solutions not only cumbersome and ineffective, but also too
expensive to implement. However, IMDEC, the most outspoken NGO against the project, released public information of
mounting costs of the Zapotillo project: the Zapotillo project's original budget (2006) was USD 750 million (USD 1,250
million in today′s value), which according to officially estimates has increased to USD 1,800 million (IMDEC, 2019).
Considering these escalating costs, the NGO argues that demand management solutions (i.e. reduction of physical losses) could
be more economical than this large infrastructure and without its large social costs.

A key anonymous actor opposing the project (personal comm. 15/05/2017) pointed out that officials of Jalisco′s water authority
are not interested in demand management strategies, because they benefit the interests of large real estate companies who need
more water rights to keep building housing developments, "it is the nature of capitalism, to keep growing […] this [the
Zapotillo conflict] is actually a class conflict."

**4.2.2 Negative consequences for the donor basin**

In the past decades Los Altos has experienced two major socio-economic changes. First, a decreasing rural population due to
migration to the United States (Durand and Arias, 2014) and to nearby cities in Jalisco. Second, the increasing industrialization
of the regional economy. In the 1990s, Mexico liberalized its markets and supported agriculture for export. These policies
helped industrialize the agricultural sector of Los Altos (Cervantes-Escoto et al., 2001). Currently, the region is the second
largest producer of animal protein in the country (Ochoa-García et al., 2014), and hosts one of the largest egg producers in the
world (WATTAgNet, 2015). This economic development has increased competition for water, especially groundwater, due to
the government′s restrictions on surface water use (DOF, 2018). Several water users confirmed the existence of a black
groundwater market, and groundwater rights grabbing in hands of industrial farmers. Consequently, most aquifers present
serious water balance deficits, which jointly amount to more than 150 hm$^3$/year in Los Altos' aquifers (CEA Jalisco, 2018);
and many have presence of selenium, fluoride and arsenic (Hurtado-Jimenez & Gardea-Torresdey, 2005, 2006). As agricultural
outputs keep increasing around 9%/year (Ochoa-García et al., 2014), groundwater overexploitation may exacerbate in the
future due to an increasing water demand. Although there are no clear numbers on the water balance for surface and
groundwater separately, water authorities calculated a combined renewable water availability in the Verde River Basin, which
also includes groundwater in Aguascalientes (Figure 1), of 1,624 hm$^3$/year, while current water demand was 1,804 hm$^3$/year
(Conagua-Semarnat, 2012).

The Observatory´s leadership has defended the interests Los Altos´ farmers by pitching the human right to food as equally important to the human right to water, which is used by Jalisco´s government. Due to the water deficit in the basin and the effects of climate change, the technical chair of the Observatory has argued that there is insufficient water in the basin to fill the dam at the planned 105 m height, and that, based on the precautionary principle, the Verde River Basin should not be burdened with additional commitments due to a water transfer. Additionally, he stated that water information provided by gauging stations in the Verde River Basin cannot be trusted, as the network of hydrological stations is allegedly defective and unattended.

An interviewee from CEA-Jalisco (personal comm. 20/04/2017) did not deny the possibility of some defective hydrological gauging stations, but claimed that even if it is true that run-off is overestimated in the basin, CEA-Jalisco is confident that the gauging station at the entry point of the dam is reliable. This station has measured an average flow of 599 hm$^3$/year (IMTA, 2015), which is enough to fill the Zapotillo dam in one year at a height of 80 m, or in two years at a height of 105 m. Currently the Verde River water flows to the Santiago River with only minor abstractions (UNOPS, 2017d). However, farmer representatives in Los Altos stated in a meeting that, even if these surface water resources of the Verde River exist (they insist that the flow of the river has dramatically decreased over the past years), these should be used to contribute to the potential growth of Los Altos.

The Jalisco´s government official addressed this continuous growth of agricultural groundwater demand as the main sustainability problem in the basin, and suggested farmers should become more efficient and stop groundwater over-exploitation (personal comm. 22/05/2017); but such an endeavor might be more complex, as described by a representative of a large industrial protein producer in Los Altos (personal comm. 02/05/2017) "[Groundwater over-exploitation] does not constrain economic development. […] If you need water you can get it in the black market. Because of corruption, Conagua cannot stop groundwater over-exploitation." The procedure to acquire or renew a groundwater right is a legal conundrum that forces farmers to hire 'coyotes' (literally: a relative of wolves, here are meant officials within Conagua that illegally ease the procedure for a considerable fee). This situation has forced smallholder farmers to sell their lands for a penny and migrate when they cannot renew their groundwater rights, since as three interviewees confirmed that "a land without water is worthless." Large producers have the means to hire coyotes and have been grabbing water rights and large portions of land from impoverished farmers.

Regarding the dam´s height and the three communities under threat of displacement, the controversy lies in incompatible values. These communities reasserted their rights of consultation and consent, participation, and the protection of their cultural and historical heritage. In turn, the government of Jalisco reasserted the utilitarian argument of the greatest good for the largest number of people. Temacapulín's representatives proposed a dam with a height of 60 m, whereby the towns would be safe from flooding. However, a smaller dam would not be able to transfer the agreed volume of water to Guadalajara and León, since the dam's storage capacity would then be 145 hm³, too small to sustain a steady water transfer of 8.6 m³/s. At a height of 80 m, Temacapulín, Acasico and Palmarejo would be flooded. However, CEA-Jalisco's representatives claimed that the construction of dikes could prevent this, albeit only for Temacapulín. IMDEC, the NGO accompanying the affected

communities, and representatives of Temacapulín are against this solution as it would create a huge unnecessary risk for the inhabitants in case the dikes fail. Moreover, an 80 m dam with a capacity of 411 hm³ would not be able to allocate sufficient water for both León and Guadalajara. With a height of 105 m and a storage capacity of 910 hm³, the dam could potentially supply sufficient water for both Guadalajara, León, and Los Altos.

**4.3 Analysis of scientific products**

The history of the conflict over the Zapotillo project has created several scientific products that have attempted to address the many uncertainties and risks of a project of this magnitude. But most of them have not analyzed the system in an integrated way. The first one (IMTA, 2005), assessed the relationship between the dam's size and its maximum water yield. Although this study explored scenarios of future water demand in the donor basin, it did not explore scenarios of the effect of climate change on precipitation, which is officially recognized as likely to decrease in Jalisco (Martínez et al., 2007). Moreover, the

study did not consider the effect of increasing groundwater over-exploitation in the basin on the base flow of the river. The study recommended the most optimistic scenario where surface water use in the donor basin would not increase in the future. Conagua (2006, 2008) subsequently released the Environmental Impact Assessment of the project, which dismissed any potential negative impact on the donor basin, based on the argument that local farmers have caused already most of the environmental degradation. However, the study analyzed the impact of the dam only at the dam site, not the overall regional

impact (CACEGIAEJ, 2018). Later, when the dam design was redesigned to 105 m in 2007, IMTA did not release any complementary study to assess the implications of a larger reservoir area, of an additional water user (Guadalajara), nor of a higher water allocation.

In 2014, the Los Altos' Animal Farmers Association commissioned ITESO (the Jesuit University in Guadalajara) to study the possible social effects of the water transfer. The study (Ochoa-García et al., 2014) concluded that according to official data the

Los Altos region already had a groundwater deficit of more than 100 hm³/year and growing, due to the continuing growth of the agricultural output of the region. It also concluded that, since the region's climate is semi-arid, the region was especially vulnerable to droughts, hence the water transfer project would have serious negative socio-economic and environmental effects. However, the study could not make a surface water assessment nor a climate change analysis due to lack of information. Recently, the Observatory made public a haphazard water footprint analysis to assess the water needed for supporting the

agricultural activity in the region (Ágora, 2018). It concluded that the water footprint of Los Altos agricultural output was 14,081 hm³/year, therefore the 12 hm³/year allocated to animal farming in the allocation agreement of the Verde River of 1997 was insufficient. However, this argumentation is flawed, since they did not consider that the water footprint of a given agricultural product includes the virtual water imported from other regions in the form of fodder. So, the actual water needed by the region is much less than 14,081 hm³/year.

To counter the study of Ochoa-García et al. (2014), and to prove that there was enough water availability in the basin, CEA-Jalisco conducted a new water availability study (IMTA, 2015). Although this time the study included climate change as a

variable in the water resources by using IPCC's regional models based on RCP-4.5 and RCP-8.5 climate scenarios, the study discarded the negative effects of climate change on the water balance due to its high uncertainty: "Climate change results should not be analyzed deterministically, but probabilistically… [we should not lose] perspective that climate change studies are still in an early stage, thus, their results cannot be taken as absolute truths, due to their low probability of occurrence… There is no certainty that projected rainfall and temperatures in climate change models will occur." (Our translation from IMTA, 2015: 212). The study did not consider possible future increases in water demand nor evaluated the dam´s behavior according to input variables (river run-off) and output variables (water allocation and other losses). As a result, the study could conclude that sufficient water was available in the Verde River Basin to comply with the water allocation agreement and environmental flows for the coming decades. The study was discredited by the leadership of the Observatory, who accused IMTA of allegedly forging data.

What can be concluded from the previous studies is that there were at least four important uncertainties that were still ignored: (1) physical groundwater processes and the interaction between groundwater and surface water in the Verde River Basin, (2) the effect of future water demand in Los Altos' water resources, (3) the effect of climate change, and (4) potential impact on water quality and ecosystem services downstream in the Santiago River. Moreover, the studies did not consider other possible alternatives to the Zapotillo project for water supply to Guadalajara and León.

As previously mentioned, in late 2014, Jalisco's government hired UNOPS to develop a comprehensive water resources model of the Verde River Basin. UNOPS' multidisciplinary team of international experts addressed the four uncertainties in the following way. 1) They analyzed groundwater dynamics by using information from NASA's GRACE earth observation project. 2) For two years, the team collected social and hydrological information in situ from the Verde River Basin to estimate current water demand and project future water demand. 3) They used IPCC's RCP-8.5 regional model of climate change for Los Altos. And 4), they calculated environmental flows downstream of the Zapotillo dam. These analyses were used as input variables for the water resources model of the Verde River Basin using WEAP software, which allowed the simulation of future scenarios (for a more detailed description of the model see supplementary material).

After months of speculation over UNOPS' results, the team released a preliminary study, which found that current water demand was 50% higher compared to official data (UNOPS, 2017c). Months later, they presented the final results in a public meeting (29 June 2017). The UNOPS team developed five main scenarios with different variables (see Figure 2). Although UNOPS' team could have developed many other scenarios with different variables, the report of the study justified choosing these five scenarios in the following way "the definition of the number of scenarios is not absolute, but may be subject to future changes at any time that it is required to attend to different questions from those raised in the framework of this study […] Specifically, it is interesting to know under which configuration of the dam´s height and volume of water transfer can guarantee [the satisfaction of] water demand and what percentage of satisfaction corresponds to it, which leads to justifying technically the presence of the dam and its geometric configuration. It is important to be clear that this focus considers only the hydrological aspects related to the satisfaction of demands. Any other conclusion about the configuration of the Zapotillo project needs to be complemented by broader technical analyses […] social and economic evaluations, among others, which

fall outside the scope of this study." (UNOPS, 2017b: 27-28). They assessed the performance of each scenario based on reliability (to supply urban water), vulnerability (volume of unmet water demand) and resilience (of the dam to recover its water levels after an empty period) indicators. The UNOPS team concluded that only scenario five scored positively on the three indicators. However, the good performance of scenario five (Figure 2) depended on reducing by 13% the volume of

water to be transferred to León, Guadalajara, and Los Altos in accordance with the 2007 agreement. The UNOPS team recommended Jalisco's government to proceed with the project with such settings and a dam height of 105 m. Jalisco's governor immediately confirmed this decision during the public presentation of the results: "We are going after the benefit of the majority and what Jalisco needs […] May history single me out for being the harbinger of the services that our people need."

The consultants immediately left the venue after the presentation, leaving no time to discuss with the attending stakeholders the key assumptions of the model, nor the justification and relevance of the five scenarios. Temacapulín's representatives reacted negatively, as their community would be flooded, and took over the podium and declared: "[The government] paid 4.6 million dollars for this stupid study, it's not a real study, it is a study of lies." (our translation). Later, Temacapulín's representatives demonstrated in front of Jalisco's government main building and declared that "We do not accept the UNOPS

team's recommendation because the decision was made beforehand […] [the UNOPS' team] did not research for alternatives, all the variables referred to the dam." (our translation).

The local academics criticized the UNOPS team's study for not considering climate change nor future water demand in scenario five, the limitations of the chosen indicators, and the still incomplete assessment of groundwater given the low reliability of GRACE's coarse spatial resolution data. Members of the Observatory interpreted these omissions in the study as

deliberate: "[T]hey applied a methodology that was biased to get the results that we heard [in the presentation]: a 105 m dam […] It makes me worried that organizations like this [UNOPS] be used to do this kind of research […] We will surely present a formal complaint in the United Nations." (this is an excerpt from a public interview with the head of the Observatory, Radio UdeG Guadalajara, 2017, our translation).

To explore the possibility of a deliberate omission, Figure 5 shows a comparison between scenario 5 and our own scenario,

which configures a scenario with the allocation variables of scenario 5 and the climate change and future water demand variables of scenario 4, as described in section 3 and illustrated in Figure 2. The results show a poor performance of the Zapotillo dam's projected storage and the three indicators chosen by UNOPS (Figure 6); whereas scenario 5 shows all three indicators (reliability, vulnerability, and resilience) on target, our scenario results into substantially lower performance, notably on vulnerability and resilience. Therefore, the poor results of these indicators do not seem to justify the implementation of the

Zapotillo project as it is currently designed.

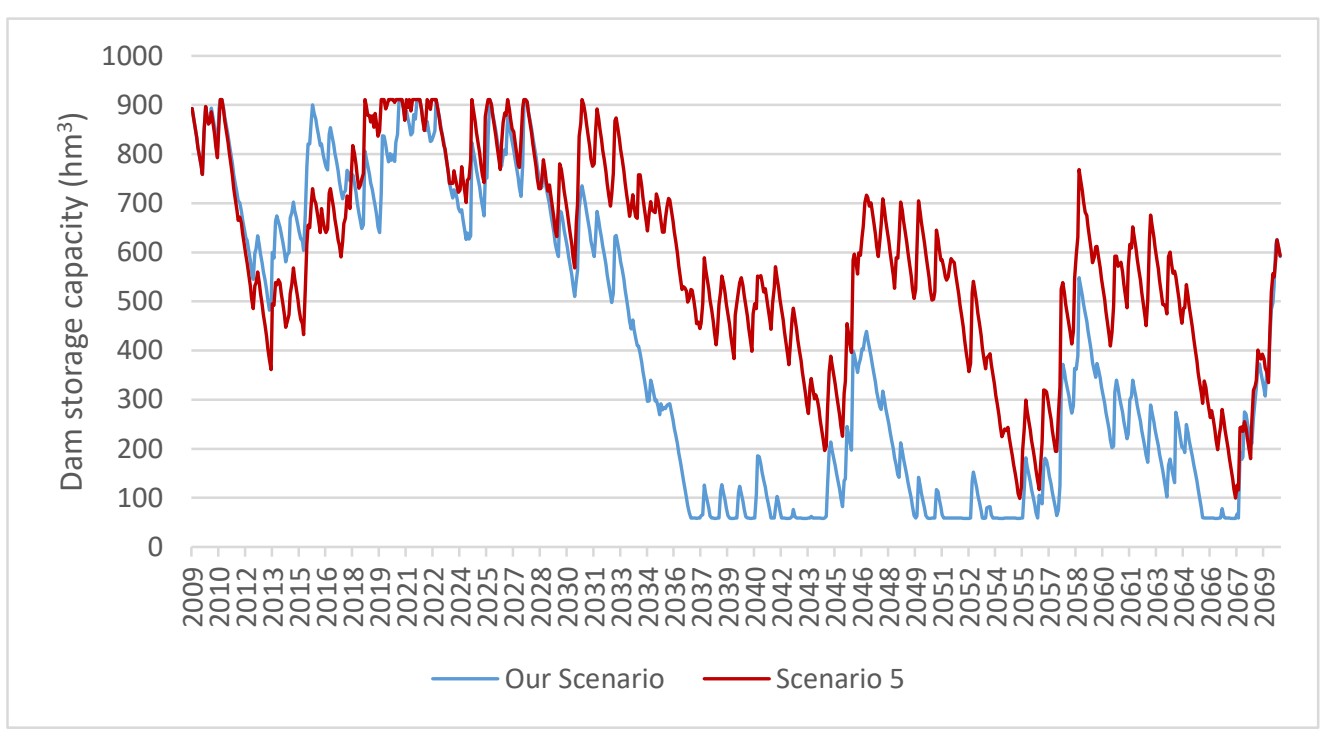

**Figure 5: Comparison of Zapotillo Dam's behavior in scenario 5 (UNOPS, 2017b) and our scenario, which includes climate change and future water demand.**

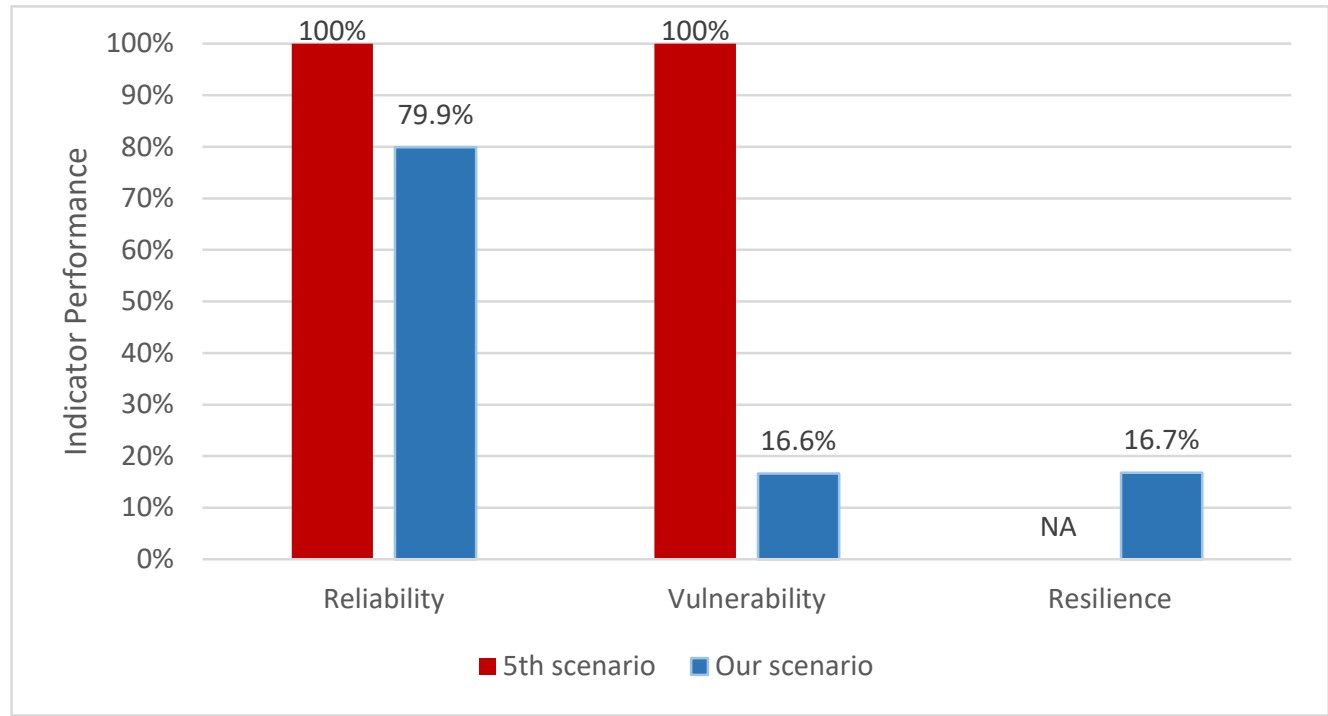

**Figure 6. Performance of the indicators for the two scenarios.[5]**

## 5 Discussion

Since large infrastructural projects are still depicted as the main solution to current water problems (Muller et al., 2015; Boelens et al., 2019), it is important to critically assess the uncertainties embedded in the scientific products that support such projects in the face of the social and environmental costs they can cause. In the case of the Zapotillo project, we found that although substantial effort had been made to reduce uncertainties, those efforts were directed towards reducing uncertainties of accuracy and precision, which partially addressed epistemic uncertainties, but not the ambiguity of multiple frames: is supply augmentation the only solution for Guadalajara and León or are there alternative solutions? Should the benefit of the majority trump the rights of a minority? The UNOPS team of experts improved the assessment of four uncertainties: climate change, future water demand, groundwater dynamics and environmental flows in the Verde River Basin. It however did not improve the understanding of the Zapotillo project's adequacy to improve the urban water problems of Guadalajara and León, nor of how and to what extent the Zapotillo project would negatively affect stakeholders in the donor region.

Regarding the efforts to reduce the four uncertainties of accuracy and precision identified in the previous section, the UNOPS study improved the knowledge of the system, but not without caveats. Since the effects of climate change depend on the

---

[5] NA (not applicable): the resilience indicator only applies when the scenario projects the water storage in the dam to reach the minimum level, impeding water supply to its users.

severity (moderate or extreme) of the chosen IPCC climate scenarios, IMTA and the water authorities seemed doubtful to

accept this uncertainty in their decision-making and removed climate change as a factor to consider when developing large hydraulic infrastructure. The water balance assessment by UNOPS (2017c) found that Conagua was underestimating water demand and revealed a serious over-exploitation of surface and groundwater in the Verde River Basin. Given the difficulty to properly estimate current water demand, future water demand became a large uncertainty. The third uncertainty is still largely unresolved: the groundwater situation in the Verde River Basin. Conagua lacks sufficient measuring infrastructure to gauge

the state of the aquifers, and there are no long-term series of groundwater levels available. Also, UNOPS's use of earth observation (GRACE) to estimate groundwater added little new information; it may even have been inappropriate, given the very coarse spatial resolution of GRACE, rendering it only suitable for very large aquifers, much larger than the Verde River Basin aquifers (Castellazzi et al., 2018; Vishwakarma et al., 2018). Finally, as all previous studies, UNOPS' study also ignored possible downstream effects of the dam beyond the city of Guadalajara and until the natural outlet of the receiving Santiago

Basin in the Pacific.

Since the UNOPS team did not address the epistemic controversies and ambiguity related to the (un)feasibility of the project, the possible alternatives for water supply in the recipient regions, the possible negative effects in the donor basin, and the injustice and unfair treatment of communities in the vicinity of the dam, the results of UNOPS' study remained contentious and mistrusted. Considering the goal of urban water security, UNOPS´ model seemed to answer the wrong research question

to address the ambiguity of the conflict: how to optimize the management and operation of the Zapotillo project to guarantee the satisfaction of water demand in Guadalajara and León. Deciding this research question was a political choice that determined the outcome of the research, since it implied that the decision to proceed with the infrastructure is already taken, and that the only valuable decision criteria are those related to optimizing the water supply to Guadalajara and León with that infrastructure, leaving other controversies described in this paper unaddressed. The reaction of actors to the UNOPS´ study is

clear; their impression is that the study and research was restricted only to the dam configuration, which was only one issue, among many, of the problem and the conflict.

The importance of asking the right question is highlighted by DFID (2013) and Feldman and Ingram (2009) who argue that the impact of research and development may decrease when it lacks a deliberative process with stakeholders, including in the definition of what the research questions are. In general and since the 1990s, research has been consistent in promoting

knowledge co-production to solve pressing and disputed environmental problems (i.e. Funtowicz & Ravetz, 1994; Van Cauwenbergh, 2008; Brugnach et al., 2011; Islam & Susskind, 2015; Armitage et al., 2015; Norström et al., 2020). The UNOPS team therefore missed the opportunity for answering a much more relevant question for all actors in the conflict: and based on decision criteria (and indicators) agreed by all stakeholders; how does the Zapotillo project compare to alternative solutions for creating a sustainable and socially just urban water system?

The knowledge generated by the UNOPS team effectively filtered out other feasible solutions to the water problems of the three regions in conflict and did not take into consideration downstream users nor environmental flows for the Santiago River. If the goal is to achieve water security and solve a water conflict, then it was not justified to restrict the research and modelling

to supply augmentation scenarios with the Zapotillo project. According to the best social and hydrological knowledge available, it can be inferred from our scenario that there are insufficient surface water resources to satisfy the demand of the three regions' explosive demographic and economic growth, which means that at least one region will continue to unsustainably deplete its groundwater resources. In fact, UNOPS fifth scenario generated positive results only because it considered null demographic and economic growth for the future and did not consider climate change in the Verde River Basin.

The case and the persistence of the conflict blocking the dam project, shows that water authorities have lost their power to impose their decisions and need the support and legitimacy of the incumbent social actors in the donor region. Given the absence of a legitimate authority to enforce decisions, actors from the three regions have entered the knowledge arena to build their cases that support their interests. Norström et al. (2020) proposed that pluralistic, goal-oriented, interactive, and context-based knowledge co-production can improve system understanding and reduce conflicts. The opposite also seems to be true - when actors in conflict produce knowledge only in relation to their interests and in isolation, they reinforce their frame and lose the overall perspective of emerging problems in the coupled water-human system at hand. In those cases, science is not able to depoliticize the conflict, but instead the conflict ends up politicizing the science-policy process. This became evident when most actors in the conflict produced or claimed unverifiable knowledge, which was never put to the test. In contexts of conflict, creating agonistic spaces to test knowledge is an important process to positively challenge knowledge claims and stakeholders´ frames (Krueger et al., 2016). However, there was a lack of systematic analysis, methodological transparency and open discussion from which firm conclusions could be drawn from the side of both the water authorities and opposing actors like the Observatory, academics, communities, and the NGOs. Especially the Observatory produced unverifiable but allegedly scientific knowledge that hardened the multiple frames at play and contributed to an increased ambiguity and partisan science.

Although the conflict is related to the control of surface water resources, groundwater seems to be a defining issue and emerging problem in the conflict. The three regions are competing for limited surface water resources aimed at protecting their available groundwater resources and their current and future demographic and economic growth. However, given the heavy reliance on groundwater for water supply, other threats seem to have been overlooked. Water quality and land subsidence has been almost absent in the debate, even though there is increasing evidence that groundwater quality is rapidly declining and land subsidence is increasing as over-exploitation intensifies (for Guadalajara see Hernández-Antonio et al., 2015; Morán-Ramírez et al., 2016; Mahlknecht et al., 2017; for León see Villalobos-Aragón et al., 2012; Cortés et al., 2015; Hoogesteger & Wester, 2017; and for Los Altos see Hurtado-Jiménez & Gardea-Torresdey, 2005, 2006, 2007).

This case study serves as a cautionary tale for actors in a water conflict, who are embroiled not in solving the problem, but in implementing their own preferred solution. Madani (2010) warned that the behavior of non-cooperative actors might result in a worse condition for all. Although science has the potential to bridge the positions of actors, it can also be misused by hegemonic actors to support their own solutions. However, as this case exemplifies, that can be counter-productive and backfire instead.

## 6 Conclusions

This paper sought to scrutinize and unravel the entanglement of politics and science in the production of water knowledge for intractable conflicts, by analyzing the case of the Zapotillo conflict in Mexico. The conflict is defined by epistemic uncertainties, ambiguity, and incompatibility of values. The first two consist of several knowledge controversies regarding water availability and the negative effects of the water transfer and dam construction in the donor basin, and the possible alternatives to supply augmentation strategies in the recipient basins. The latter consists of a dispute over the distribution of the environmental, social, and economic costs and benefits derived of the Zapotillo project.

This study has two main findings. 1) Intractable water conflicts tend to isolate the process of knowledge production, which foregrounds issues that are politically convenient for each actor, while other issues, perhaps more important for sustainability (like groundwater over-exploitation) are concealed and remain unaddressed. And, 2) isolated knowledge has less potential for transforming the conflict by missing core epistemic uncertainties and pushing value-laden knowledge claims as facts. After analyzing the model of UNOPS, we found that its research team made a significant contribution to knowledge by reducing uncertainties related to precision and accuracy of future water demand, climate change, groundwater dynamics and ecological flow. But the team failed to address epistemic uncertainty around emerging problems induced by groundwater over-exploitation as well as ambiguity related to the negative effects in the donor basin and more sustainable and socially just alternatives to the Zapotillo project. We found some indications that the UNOPS team indulged into what Boelens et al. (2019) call the manufacture of ignorance, by recommending Jalisco's government to build a 105 m dam without taking into account climate change, future water demand, nor alternative water supply options. But this result may also be explained by the absence of efforts by the UNOPS team to facilitate the co-production of knowledge. So, even if the UNOPS team did not deliberately indulge in the manufacture of ignorance by building a water resources model based on political interests, its research suffered from tunnel vision by inadequately managing the ambiguity of the conflict. Nevertheless, the mere suspicion of deliberate manufacture of ignorance was enough to discredit UNOPS results by most stakeholders. However, contrary to the conclusion of Boelens et al. (2019), deliberate production of biased knowledge is not exclusive to powerful actors. Instead, this kind of knowledge was produced by most of the actors in the conflict.

Returning to the original question whether science can depoliticize conflicts or whether science is politicized in the process, this case has shown that attempting to depoliticize science-policy processes is very difficult, since these processes are inherently political. Moreover, involving alleged neutral - or apolitical - third parties to depoliticize scientific knowledge to resolve water conflicts can backfire if they act - or are perceived - as stealth advocates of political interests. However, we identified two elements that can contribute to a possible transformation of the conflict and management of such politicization. First, scientists in contexts of conflict should be aware of not promoting specific solutions, since that is the role of the political actors. When scientists assume the role of "honest broker of policy alternatives" (Pielke, 2007), it restrains them from offering a specific course of action and compels them to expand the scope of choice for the actors in the conflict. And second, to

promote social mechanisms to filter as much as possible which knowledge claims are more value-laden, and which are less so, particularly in contexts of conflict and high uncertainties. There is an urgent need to design water resources models in a more open way to allow the participation of stakeholders and legitimize the data used in them (Islam & Susskind, 2018) as well as the values hidden in them; this can support the necessary task of reviewing alternatives to large infrastructures (Van der Zaag & Gupta, 2008). Additionally, fostering stakeholder participation could collaboratively bring about socially relevant research questions that open the decision space (Voinov & Gaddis, 2008; Zimmerer, 2008; Budds, 2009; Lejano & Ingram, 2009; Brugnach et al., 2011; Blöschl et al., 2013; Armitage et al., 2015; Basco-Carrera et al., 2017; Van Cauwenbergh et al., 2018; van der Molen, 2018; Norstöm et al., 2020). Brugnach et al. (2011) support this as one of the main strategies to handle ambiguity, albeit with the drawback of necessary high social skills to bring people together, which, in a context of conflict, is difficult to achieve. However, despite this difficulty, attempting such an effort could already improve the capacity to innovate by incorporating new perspectives, as suggested by Brugnach et al. (2008), and by identifying arbitrary decisions in public policies by hegemonic actors. Such transparency could decrease the capacity of powerful actors to capture the science-policy process. However, further research is needed to evaluate if co-production of knowledge can bring about cooperation and consensus between the stakeholders and limit the influence of politics and vested interests in decision-making in water conflicts.

*Data availability.* The reader can access the Verde River Basin model developed by the UNOPS team of experts and modified by the authors at: https://github.com/jongmadrigal/Verde-River-Basin. Although the model is only accessible through the software WEAP (www.weap21.org), it is possible to download the software for free and run its test version to replicate this article´s findings.

*Author contribution.* Conceptualization, JGM, NVC and PvdZ; Data curation, JGM; Formal analysis, JGM; Investigation, JGM and NVC; Methodology, JGM, NVC and PvdZ; Software, JGM; Supervision, NVC and PvdZ; Writing—original draft, JGM; Writing—review & editing, NVC and PvdZ.

*Competing interests*. The authors declare that they have no conflict of interest.

*Acknowledgements*. We would like to thank the two reviewers and the editor for their useful comments on this paper. This research could not have been possible without the support of CONACYT, the Mexican Council for Science and Technology.

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
