# Peer review of "Unravelling intractable water conflicts: the entanglement of science and politics in decision-making on large hydraulic infrastructure"

_Hydrology and Earth System Sciences, 2020_

## Referee Comment (RC1) · Jeroen Vos (Referee) · 29 Mar 2020

Dear Editor,

The paper on the El Zapotillo dam project in Mexico is a relevant and well-explained example of the political use of a hydrological model.

The article shows how the hydrological model, elaborated by UNOPS and commissioned by the government of Jalisco (recipient basin) did not take into account: (a) important negative effects for the to-be-displaces communities, ecosystems and donor-
basin farmers; (b) important factors like climate change, increased demand in the cities, water quality, aquifer depletion and land subsidence; and (c) important potential alternative solutions like demand management in the cities (including reduction of distribution losses) and local water supply options.

Conceptual framework The conceptual framework does draw on relevant literature and specifies the different relevant problems related to the construction and use of hydrological models. However, regarding the political nature of models the main concepts introduced are the "epistemic uncertainty and ambiguity" (lines 71-77). I think it is much clearer to see the "epistemic uncertainties and ambiguities" as political choices of the modellers. The variables in the model reflect the value the factors have for the modellers. Not taking into account the ecological status of the downstream river, or the historic, religious and cultural value of the church in Temacapulín are political choices. This does not become clear by calling that choice an "epistemic uncertainty, understood as the ignorance of the functioning of a given system", or an "ambiguity, understood as multiple knowledge frames" (line 73). Those choices are not so much an issue of understanding the functioning or having the right knowledge: it is instead about political choices (based on interests, valuing and worldviews).

Method It would be good to include better scrutiny of the position and work procedures of UNOPS. As part of the methodology is would have been interesting to collect data on, and do interviews with, the involved experts from UNOPS. Why did they make the model? Why did they not take into account the actors, factors and alternative solutions? Were they aware of their political role (or did they believe themselves in their political neutrality)? Furthermore, it does not become clear what role the UNOPS model outcomes have played in the development of the controversies over the El Zapotillo project. It would have been interesting to research more on how the different stakeholders have used, and reacted to the report of the model outcomes (as set out in lines 216-217). This could give a much more detail description than the general "many actors were negatively surprised" in line 227-8 and the brief description of responses

of Temacapulin's representatives and academics in lines 388-393.

Argumentation In general, the argumentation is fair. The Figures 3 and 5 need more explanation. In Fig 3 the allocated flows do not coincide with the allocated flows mentioned in lines 222-223.

The Conclusions assert that co-production of more transparent and trusted knowledge will help to resolve conflicts. This might be true to some extent, but interests of stakeholders depend on (geographical, socio-economic and institutional) positions and valuing of certain costs and benefits more than others. More and better trusted knowledge will not overcome these differences.

Several typos, missing words and awkward wordings could be corrected in a thorough proof-read.

---

## Author Comment (AC1) · 3 Apr 2020

Dear Jeroen Vos,

We thank you for your comments, we found them insightful and they will certainly help to improve the quality of the paper.

Conceptual framework. Regarding your first comment on the conceptual framework: "I think it is much clearer to see the "epistemic uncertainties and ambiguities" as political choices of the modellers. The variables in the model reflect the value the factors have

for the modellers. Not taking into account the ecological status of the downstream river, or the historic, religious and cultural value of the church in Temacapulín are political choices. This does not become clear by calling that choice an "epistemic uncertainty, understood as the ignorance of the functioning of a given system", or an "ambiguity, understood as multiple knowledge frames" (line 73). Those choices are not so much an issue of understanding the functioning or having the right knowledge: it is instead about political choices (based on interests, valuing and worldviews)."

Answer: We believe that the main question here is whether the technical choices made by the modelers were politically motivated or were a consequence of not managing or understanding ambiguity properly, including that of the modelers themselves. The reason we prefer to discuss ambiguity instead of referring to political choice directly is to remain neutral about the level of conscious intent behind the choices made in the model. The simultaneous presence of multiple valid, and sometimes conflicting, ways of framing a problem that constitutes ambiguity is in essence a result of differences in values and worldviews. As such, it strongly relates to the appearance of political choices, whether consciously based on ambiguity or not. However, to answer this question we would need to interview the modelers; and, as explained in the answer for your following comment, that was not possible. Therefore, we considered both hypothesis, as shown in lines 484-485: "So, even if UNOPS did not deliberately indulge in the manufacture of ignorance, their research suffered from tunnel vision."

If politically motivated, the technical choices would corroborate Boelens′ hypothesis about the Manufacture of ignorance, which is described in the conceptual framework and analyzed in the discussion and conclusion of the paper. However, UNOPS′ reports and the model itself indicate that, to a certain extent, the modelers took into account environmental flows and the interests of Temacapulín. They evaluated environmental flows immediately after the Zapotillo dam, but dismissed the ecological contribution of the Verde River to the Santiago River, as mentioned in line 441. And then, considered one scenario where Temacapulín would not necessarily need to be relocated (i.e. third

scenario in Figure 3, with a dam of only 80 meters height). The concepts of epistemic uncertainty and ambiguity helped explain how UNOPS determined the system boundaries that delimited their model (e.g. that the environmental flows for the Santiago river were outside the scope of their objective, which was to study the Verde River Basin; note that UNOPS was not the first to omit the Santiago river; also for example Wester did so in his comprehensive PhD thesis on the Lerma-Chapala basin); and how UNOPS framed in their model the interests of Temacapulín, by considering one scenario where the communities are not necessarily flooded, and later discarding this scenario based on the obvious conclusion that it would not fulfill the accorded water allocation to the three regions. As previously said, it is difficult to understand the motivations of these technical decisions without interviewing the modelers in depth. We will change the wording of lines 484-485 to make this analysis clearer: "So, even if UNOPS did not deliberately indulge in the manufacture of ignorance by building a water resources model based on political interests, their research suffered from tunnel vision by inadequately managing the ambiguity of the conflict."

Method. On your first comment about method: "It would be good to include better scrutiny of the position and work procedures of UNOPS. As part of the methodology is would have been interesting to collect data on, and do interviews with, the involved experts from UNOPS. Why did they make the model? Why did they not take into account the actors, factors and alternative solutions? Were they aware of their political role (or did they believe themselves in their political neutrality)?"

Answer: We actually had the same questions. We requested numerous times meetings and interviews with the modelers during summer of 2017, when the entire UNOPS team was stationed in Mexico. Our requests, along with other researchers from local universities, were ignored. Since the modelers were not Mexicans, once they delivered and presented the results, they all went back to their countries of origin. We continued to request for interviews, albeit electronic ones, and we were still ignored. Therefore, the answers to all those pertinent questions can only be inferred; but we preferred not

to speculate about them in the paper.

On your second comment about the method: "Furthermore, it does not become clear what role the UNOPS model outcomes have played in the development of the controversies over the El Zapotillo project. It would have been interesting to research more on how the different stakeholders have used, and reacted to the report of the model outcomes (as set out in lines 216-217). This could give a much more detail description than the general "many actors were negatively surprised" in line 227-8 and the brief description of responses of Temacapulin's representatives and academics in lines 388-393."

Answer: That we find an excellent suggestion. Many actors released public statements, which are available online, condemning the results of UNOPS. We will therefore include the most relevant statements in the revised version of the paper.

Argumentation. Regarding the first comment on argumentation: "The Figures 3 and 5 need more explanation. In Fig 3 the allocated flows do not coincide with the allocated flows mentioned in lines 222-223."

Answer: The allocated flows mentioned in lines 222-223 are: 8.6 m3 /s, 4.8 m3 /s and 7.5 m3/s. These flows represent the aggregated flow of water to be allocated by the dam to the three regions, for three kind of scenarios (the first value refers to scenario 1 and 2, the second to scenario 3, and the last one to scenario 5). Figure 3 shows the disaggregated allocated flows to each region. In the revised version we will modify the text to make this clearer.

On the second comment about the argumentation: "The Conclusions assert that co-production of more transparent and trusted knowledge will help to resolve conflicts. This might be true to some extent, but interests of stakeholders depend on (geographical, socio-economic and institutional) positions and valuing of certain costs and benefits more than others. More and better trusted knowledge will not overcome these differences."

Answer: In lines 500-501 we acknowledge the limitations of co-production of knowledge: "This might not be a panacea against vested interests (Molle, 2008), but can be an improvement to identify arbitrary decisions in public policies by hegemonic actors." We recognize that although irreconcilable interests and positions maybe inherent to water conflicts, co-production of knowledge can limit the instrumental use of knowledge as a hegemonic tool against non-technical actors, which is a step forward to fairer conflict resolutions. In addition, we will clarify the importance of managing the co-creation of knowledge with appropriate methods, acknowledging the potential backfiring of more knowledge per se, in reference to Dewulf et al. 2011.

On the third comment about the argumentation: "Several typos, missing words and awkward wordings could be corrected in a thorough proof-read."

Answer: We will revise the paper again to correct the typos and missing words in the revised version of the paper.

Best regards,

Jonatan, Nora and Pieter.

––––––––––––––––––––––––––

---

## Referee Comment (RC2) · Jeroen Vos (Referee) · 8 Apr 2020

Dear authors,

Thank you for your reply to my comments. The reply gives thorough clarifications, and also explains the way in which you will clarify and improve the manuscript.

I agree with all but one point, and that is the political neutrality of the model. In my view the model is political as it influences decision making in a biased way (which has consequences for the distribution of costs, benefits and risks among different social groups

and the environment). It does not matter if these consequences are intentional or not. One could compare with the following example: A member in an appointment selection committee that is unconsciously racist and has a negative bias against coloured candidates will make the selection procedure racist and the outcomes biased towards white candidates. Even if in an interview this member states not to be a racist and did not have racist intentions. (Similar cases are the male-biases in appointments of CEOs).

In the same vein the model is political, even if the makers did not intent it to be political. In the case of the El Zapotillo dam project, you say that the modellers took into account the interests of the Temacapulín community by including an option of 80 m dam height. However, with 80 m the communities of Acasico and Palmarejo will be flooded, and Temacapulín will have to be protected with dykes. Those communities prefer a dam height of 40 m not 80. Many choices are made in including or excluding interests, people and values. The UNOPS model also neglected the effect of climate change and future increase of water demand. This makes the model political in the sense that the numerical objectifications such as dam height, rankings and indicators influence political options and imaginaries, resulting in biased discussions and instruments of governance that influence decision makers (but also other stakeholders like academics, NGOs and the involved communities).

Ambiguity is something different. Ambiguity is the fact that you can look at the same object in different ways and thus have different conceptions of what it is. The UNOPS model is not ambiguous in that sense. The model promotes a selected number of problem definitions, numerical objectifications, indicators and solutions and selects one preferred option with a number of (biased, not neutral) indicators. In this selection process incommensurable values are made commensurable, which implies political decisions.

---

## Referee Comment (RC3) · Leticia Lima (Referee) · 9 Apr 2020

Dear Editor and Authors,

After carefully reading the manuscript, I decided to provide my comments in this review following the questions that are used to guide reviewers provided by HESS editorial team in their website.

I believe the manuscript provides very thoughtful insights about a much needed topic and it is worth publishing. However, the authors would need to solve some particular

issues beforehand. Please consider that the following comments are entirely based on my personal experience and, therefore, they may be contested as well.

> Does the paper address relevant scientific questions within the scope of HESS?

The paper addresses an important topic which is: the causes of potential failure in science-policy processes behind fundamental decisions regarding water resources, particularly in the context of conflicts over water supply/demand and large infrastructures.

The content relates to the scope of HESS as it refers to a study of interactions of hydrological systems and human activities, focusing on water management. By being an interdisciplinary study, with a fundamental intake from Political Ecology, it also falls within one of the aims of the journal, which is the cross-fertilization across disciplinary boundaries.

> Does the paper present novel concepts, ideas, tools, or data?

There is no novel concept, idea or tools presented in the paper. However, the authors' contribution to the scientific debates around the science-policy interface and its impacts on society is noteworthy and it is accomplished by means of a thoroughly presented case study. In addition, the authors provided new data produced by means of their own modelling scenario for the case study.

> Are substantial conclusions reached?

The paper reinforces some important perspectives over the science production process and its appropriation by different actors in society. There is an important contribution to the debate on the use of science by hegemonic actors in policy processes. Particularly, it concludes emphasizing the importance of proper social participation and collaborative paths into the scientific process in order to reach better decisions when it comes to water management. However,

POINT 1) the study lacks considerations about the potential challenges and/or pitfalls of

an inclusive process in science. Although addressing these does not constitute the aim of the paper per se, I believe that at least some considerations and some references on this point should have been made. The only point on that made by the authors is: "this might not be a panacea against vested interests".

POINT 2) The conclusion section in this paper is, by far, the longest I've ever seen. There seems to be repetition of some results in it too. I believe that the conclusion section should not repeat results, but instead, provide a broader perspective on the topic raised based on the insights of the study.

> Are the scientific methods and assumptions valid and clearly outlined?

The methods section provides useful information to understand the investigation process taken by the authors. However, it is quite confusing the way the whole paper is structured and how information is provided in each section. Taking this into account, I have some points to make and some suggestions that may help the authors:

POINT 3) There are plenty of information about the case study in several different parts of the manuscript. Complementary information about the history of the case appears not only in the section called "case study" but also in the "methods", and then new information is added to "controversies", and then to "analysis of scientific products". Because of that, a large portion of the manuscript content may fall into what we could call the description of "a case study". Therefore, I believe the authors should: (A) replace the name of the section "case study" to "study area" and limit the content of this section to a very direct and precise description of the two cities, only bringing basic information such as location, demography, basic biophysical data, basic water demand numbers; (B) save the remaining content previously placed in "case study" to a new section that describes the entire history of the case, highlighting the most important facts and their context (it could go as sub-section of Results considering that it is a product of the authors' interviews), refraining to evaluate or take any stance on the case itself in this part; (C) check if there is no information in the following sections that

would be better placed in this new section dedicated to the history of the case and move it; (D) remove any repetition in the text (as an example, line 366 is a repetition); (E) please provide a STORYLINE, that is, a diagram with the most important facts so that it will help readers not to lose themselves while following this long story.

POINT 4) It is of high value the fact that the authors reproduced the modelling exercise made by a third part under Jalisco's government request with an additional scenario produced by themselves, including important issues like climate change, etc. This highlights the outstanding ability of the authors to work with interdisciplinary methods. However, it seems to me that the "methods" section is missing important components that are required when we are talking about models and simulations: there is no single paragraph describing the model in short terms and its assumptions, there is not a single table showing the main biophysical parameters used in the model, nor even a list of the data sources. I am aware that the authors are providing a link to a web page with the model and everything needed to reproduce the results, but still, I think that a basic overview of the simulation should be provided in plain text explained by the authors with additional figures+tables, including equations that describe how those indicators are calculated, even if all of that comes in the form of a Supplementary Material. Not all readers will have the required time to install and run the WEAP model in their computers to get a perspective of what this is about.

POINT 5) I truly believe that all scientists take some sort of political stance in their own topic of research. I am not against that because it seems natural to me as we are all human beings and many of us understand our role as citizens. However, I believe that scientific writing, when it comes to the description of case studies, should be done in a way that helps the readers to understand the whole story BUT without inducing them to take a side/stance by mixing description of the case with the analysis made by the authors. That said, I want to give an example taken from line 206: "... we adopted an interdisciplinary method to assess the scientific products that were developed with the intention to have a decisive role in de-politicizing the conflict, ...". There is no

undoubtful evidence in this paper that the "intention" was to "de-politicize the conflict". There is SOME evidence that it might be the case, or that it may be interpreted that way. But this sort of affirmation should be placed somewhere in the discussion section, where the authors will interpret what they've got from fieldwork based on the literature and in their own points of view. This type of phrase should not be placed in the methods section, in my opinion.

POINT 6) According to the text, the authors spent 6 months in Guadalajara, 2017, conducting in-depth interviews, including to members of the Jalisco's government. I could not find information about interviews made with members of the Guanajuato's government. Also, by being a "tale of two cities", it seems that León did not receive a visit. Is that correct? How can the authors justify that?

POINT 7) Although Figure 3 brings comparison of the five scenarios developed by a third part, which could fit into methods section, it also included the scenario developed by the authors themselves. Because of that, Figure 3 now seems more appropriate for the Results section because produced data is being brought to the study.

POINT 8) I believe the authors should have mentioned in the methods section what the specific methodology was followed to analyse and interpret the contents obtained from the interviews. One cannot know if the authors transcribed their interviews, or if they analysed data directly from audio files or notes from the field. Readers would not able to identify if the authors used some sort of technique to code the material or applied a specific method from the literature to come up with the Table 1 "controversies and frames". It would be good to know what the authors are calling controversies and frames here. P.S.: Line 216: 'to record' how?

POINT 9) Small typo in footnote 3 of page 10: If is written with capital I in the middle of the phrase.

> Are the results sufficient to support the interpretations and conclusions?

Results section brings a lot of information drawn from interviews. The authors assume a form of storytelling in order to bring the details about the controversies, mainly following the major events that happened. The results presented do support in part their interpretations and major conclusions. For instance, it seems clear that a lack of transparency and public participation during the modelling process requested by the Jalisco's government resulted in scenarios that were not entirely useful for the case, considering the lack of some aspects such as future water demands and climate change in the scenario 5, as an example. That leads to the conclusion that more transparency and participation are needed to solve such intractable problems. However, some points must be made about the results section:

POINT 10) As it seems to be the case in the whole text, there is some sort of mixing of information in the results section. I truly believe that the authors could benefit from taking a time to reorder the text, choosing more carefully where each piece of information will be placed, and maybe using more sub-sections to sort it out. They can take advantage of tables to summarize information too. Here, a question remains to the authors: how can you present the story behind the contradictions without making your reader feel lost?

POINT 11) Lines 362-365: The authors are calling the groundwater in the basin "an uncertainty". I think they should consider wording here... the groundwater cannot be called an uncertainty, in my opinion. You may consider rephrasing it to something like "the epistemic uncertainty related to the groundwater physical processes in the basin".

POINT 12) Lines 421-423: authors should provide scientific references here to support their argument that the GRACE data is not suitable in this case because of its resolution.

> Is the description of experiments and calculations sufficiently complete and precise to allow their reproduction by fellow scientists (traceability of results)?

The authors declared having all required materials and model available online to run

the reported simulations. In fact, their web link is working well, although I did not have the time to download it and try it by myself. However, as I commented previously, the methods section lacks details about model description, table of main parameters, source information about the dataset used to run the simulations, etc.

> Do the authors give proper credit to related work and clearly indicate their own new/original contribution?

Yes.

> Does the title clearly reflect the contents of the paper?

Yes. Totally.

> Does the abstract provide a concise and complete summary?

Although the abstract is well written and summarizes very well the content of the paper, I would argue that it is too long for an abstract and the authors would benefit from writing everything in the paper (not only the abstract) more concisely and direct to the point.

Tiny detail - Line 2: nobody "owns" a model in your case. A freeware was used to run the simulations. You could say, instead, that the modelling process was dominated by a particular group.

> Is the overall presentation well-structured and clear?

In my opinion, although the paper presents a very interesting case and it is worth publishing, the major drawback here is, in fact, presentation and structure. The text is too dense, it repeats in some parts, information is not well distributed, nor a good use of structures (such as tables/figures/diagrams) was made to summarize and facilitate the reading. I strongly advise the authors to consider improving the structure and readability of the paper.

> Is the language fluent and precise?

The language is fluent and there are just a few typos. Every piece of information is well understood. However, again, the writing should be straight to the point in order to improve readability.

> Are mathematical formulae, symbols, abbreviations, and units correctly defined and used?

Yes.

> Should any parts of the paper (text, formulae, figures, tables) be clarified, reduced, combined, or eliminated?

In my comments above I emphasized the need to reduce, clarify, and reorder the text into more meaningful sections. Particularly in terms of reducing the text, I would advise the authors to apply this to the abstract and the conclusion sections.

> Are the number and quality of references appropriate?

Yes.

> Is the amount and quality of supplementary material appropriate?

According to my comments about the methods section, additional information about the model, parameters and data sources could be provided in the form of a supplementary material.

---

## Author Comment (AC2) · 3 May 2020

Dear Jeroen Vos,

Thank you for this additional comment and the fruitful discussion it originates. We do not deny the influence of politics in the technical decisions of the model nor suggest that the model is politically neutral; in fact, the title makes explicit reference of precisely that: "the entanglement of science and politics in decision-making on a large hydraulic infrastructure project". However, we think that the concepts of epistemic uncertainties

and ambiguity are analytical concepts that help understand the many ways in which important decisions over infrastructure projects can be inconclusive and/or biased, be that politically motivated or not (although in this case the evidence strongly suggests that it is so). However, we still want to make the strong case for the use of these analytical concepts. Even if the technical decisions over the water resources model were made in a way that would satisfy all the interests of all the actors in the conflict, decisions made with high epistemic uncertainty over the functioning of the water system could bring unsustainable results over time (e.g. if water demand or groundwater over-exploitation rate are improperly represented). So, in that sense, the importance of good models and management of uncertainties transcends the political sphere.

Over the discussion on the concept of ambiguity, we did not apply that concept to the model, but to the socio-hydrological reality the model was trying to analyze and understand. As mentioned before, we agree that in this case the technical decisions behind the model were, to an extent, influenced by politics. However, the socio-hydrological reality is undoubtedly ambiguous, since it is extremely difficult to agree among the stakeholders which is the optimal solution to a complex problem or even to agree on what the problem really is. Thus, the concept of ambiguity is a useful concept to analyze the perspectives of all the different actors in the conflict and the irreconcilable solutions each one proposes. This allows, in our view, a much richer interplay between science and politics. If political interests are more transparent and not disguised as unbiased scientific facts, stakeholders, with the aid of science, can create more innovative and hopefully fairer solutions.

Best regards,

Jonatan, Nora and Pieter.

---

## Author Comment (AC3) · 3 May 2020

Dear Leticia Lima,

We thank Leticia Lima for the detailed and constructive comments and suggestions. To facilitate the reading of our response, we reproduce your comment verbatim and divide it into sections, then, we respond immediately after each of the points that were raised.

"Dear Editor and Authors, After carefully reading the manuscript, I decided to provide my comments in this review following the questions that are used to guide reviewers

provided by HESS editorial team in their website. I believe the manuscript provides very thoughtful insights about a much needed topic and it is worth publishing. However, the authors would need to solve some particular issues beforehand. Please consider that the following comments are entirely based on my personal experience and, therefore, they may be contested as well."

> Does the paper address relevant scientific questions within the scope of HESS?

"The paper addresses an important topic which is: the causes of potential failure in science-policy processes behind fundamental decisions regarding water resources, particularly in the context of conflicts over water supply/demand and large infrastructures. The content relates to the scope of HESS as it refers to a study of interactions of hydrological systems and human activities, focusing on water management. By being an interdisciplinary study, with a fundamental intake from Political Ecology, it also falls within one of the aims of the journal, which is the cross-fertilization across disciplinary boundaries."

> Does the paper present novel concepts, ideas, tools, or data?

"There is no novel concept, idea or tools presented in the paper. However, the authors' contribution to the scientific debates around the science-policy interface and its impacts on society is noteworthy and it is accomplished by means of a thoroughly presented case study. In addition, the authors provided new data produced by means of their own modelling scenario for the case study."

> Are substantial conclusions reached?

"The paper reinforces some important perspectives over the science production process and its appropriation by different actors in society. There is an important contribution to the debate on the use of science by hegemonic actors in policy processes. Particularly, it concludes emphasizing the importance of proper social participation and collaborative paths into the scientific process in order to reach better decisions when it

comes to water management. However, POINT 1) the study lacks considerations about the potential challenges and/or pitfalls of an inclusive process in science. Although addressing these does not constitute the aim of the paper per se, I believe that at least some considerations and some references on this point should have been made. The only point on that made by the authors is: "this might not be a panacea against vested interests". POINT 2) The conclusion section in this paper is, by far, the longest I've ever seen. There seems to be repetition of some results in it too. I believe that the conclusion section should not repeat results, but instead, provide a broader perspective on the topic raised based on the insights of the study."

Response Point 1: We consider this a valid point. However, as you mentioned, since this is not the aim of the paper, considering potential challenges and pitfalls would require future research based on experimentation with participatory approaches in water conflicts. We will add to the text some sentences for future research, as well as finding critical analysis in the literature.

Response Point 2: We will shorten the conclusion in the revised version of the paper.

> Are the scientific methods and assumptions valid and clearly outlined?

"The methods section provides useful information to understand the investigation process taken by the authors. However, it is quite confusing the way the whole paper is structured and how information is provided in each section. Taking this into account, I have some points to make and some suggestions that may help the authors: POINT 3) There are plenty of information about the case study in several different parts of the manuscript. Complementary information about the history of the case appears not only in the section called "case study" but also in the "methods", and then new information is added to "controversies", and then to "analysis of scientific products". Because of that, a large portion of the manuscript content may fall into what we could call the description of "a case study". Therefore, I believe the authors should: (A) replace the name of the section "case study" to "study area" and limit the content of this section to

a very direct and precise description of the two cities, only bringing basic information such as location, demography, basic biophysical data, basic water demand numbers; (B) save the remaining content previously placed in "case study" to a new section that describes the entire history of the case, highlighting the most important facts and their context (it could go as sub-section of Results considering that it is a product of the authors' interviews), refraining to evaluate or take any stance on the case itself in this part; (C) check if there is no information in the following sections that would be better placed in this new section dedicated to the history of the case and move it; (D) remove any repetition in the text (as an example, line 366 is a repetition); (E) please provide a STORYLINE, that is, a diagram with the most important facts so that it will help readers not to lose themselves while following this long story. POINT 4) It is of high value the fact that the authors reproduced the modelling exercise made by a third part under Jalisco's government request with an additional scenario produced by themselves, including important issues like climate change, etc. This highlights the outstanding ability of the authors to work with interdisciplinary methods. However, it seems to me that the "methods" section is missing important components that are required when we are talking about models and simulations: there is no single paragraph describing the model in short terms and its assumptions, there is not a single table showing the main biophysical parameters used in the model, nor even a list of the data sources. I am aware that the authors are providing a link to a web page with the model and everything needed to reproduce the results, but still, I think that a basic overview of the simulation should be provided in plain text explained by the authors with additional figures+tables, including equations that describe how those indicators are calculated, even if all of that comes in the form of a Supplementary Material. Not all readers will have the required time to install and run the WEAP model in their computers to get a perspective of what this is about. POINT 5) I truly believe that all scientists take some sort of political stance in their own topic of research. I am not against that because it seems natural to me as we are all human beings and many of us understand our role as citizens. However, I believe that scientific writing, when it comes to the description of

case studies, should be done in a way that helps the readers to understand the whole story BUT without inducing them to take a side/stance by mixing description of the case with the analysis made by the authors. That said, I want to give an example taken from line 206: ". . . we adopted an interdisciplinary method to assess the scientific products that were developed with the intention to have a decisive role in de-politicizing the conflict, . . .". There is no undoubtful evidence in this paper that the "intention" was to "de-politicize the conflict". There is SOME evidence that it might be the case, or that it may be interpreted that way. But this sort of affirmation should be placed somewhere in the discussion section, where the authors will interpret what they've got from fieldwork based on the literature and in their own points of view. This type of phrase should not be placed in the methods section, in my opinion. POINT 6) According to the text, the authors spent 6 months in Guadalajara, 2017, conducting in-depth interviews, including to members of the Jalisco's government. I could not find information about interviews made with members of the Guanajuato's government. Also, by being a "tale of two cities", it seems that León did not receive a visit. Is that correct? How can the authors justify that? POINT 7) Although Figure 3 brings comparison of the five scenarios developed by a third part, which could fit into methods section, it also included the scenario developed by the authors themselves. Because of that, Figure 3 now seems more appropriate for the Results section because produced data is being brought to the study. POINT 8) I believe the authors should have mentioned in the methods section what the specific methodology was followed to analyse and interpret the contents obtained from the interviews. One cannot know if the authors transcribed their interviews, or if they analysed data directly from audio files or notes from the field. Readers would not able to identify if the authors used some sort of technique to code the material or applied a specific method from the literature to come up with the Table 1 "controversies and frames". It would be good to know what the authors are calling controversies and frames here. P.S.: Line 216: 'to record' how? POINT 9) Small typo in footnote 3 of page 10: If is written with capital I in the middle of the phrase."

Response Point 3A & 3B: We think this may provide more clarity to the paper, so we

will split the case study section into study area and case study in the results section in the revised version.

Response Point 3C-3E: We will revise again the story of the conflict to make it clearer, and create a figure showing the storyline of the conflict.

Response Point 4: We will provide an additional table in the supplementary material summarizing the main components of the model.

Response Point 5: We will delete this phrase from the method section and add it in the discussion section.

Response Point 6: No, we did not interview stakeholders in León. We have three reasons for that: one, since the negative consequences from the Zapotillo project would be carried in Jalisco, the central scientific controversies, and the focus of our research was also here; two, all the actors that we interviewed had a deep knowledge with the relationship with the city of León; and three, we complemented this knowledge through relevant literature on the issue from the perspective of León, as well as relevant official documents from its main stakeholder, SAPAL, the water utility of León. We are confident that our identification of Leon's role in the conflict through interviewing Jalisco's actors and mobilizing secondary data is sufficient to support the analysis of the case. We are adding a sentence in the methodology explaining this.

Response Point 7: We respectfully disagree with this point. Figure 3 should be part of the methods section because it only indicates the parameters that we introduced in the model for a new scenario. The results are described in Figure 4.

Response Point 8: We will provide additional information on this in the revised version of the paper. We electronically recorded some interviews and later transcribed them; and some other interviews, due to the delicate nature of the conflict, were not recorded, only chronicled through fieldnotes. The informal interviews mentioned in line 216 were not electronically recorded, but only chronicled in fieldnotes.

Response Point 9: We will correct that typo in the revised version of the paper.

> Are the results sufficient to support the interpretations and conclusions?

"Results section brings a lot of information drawn from interviews. The authors assume a form of storytelling in order to bring the details about the controversies, mainly following the major events that happened. The results presented do support in part their interpretations and major conclusions. For instance, it seems clear that a lack of transparency and public participation during the modelling process requested by the Jalisco's government resulted in scenarios that were not entirely useful for the case, considering the lack of some aspects such as future water demands and climate change in the scenario 5, as an example. That leads to the conclusion that more transparency and participation are needed to solve such intractable problems. However, some points must be made about the results section: POINT 10) As it seems to be the case in the whole text, there is some sort of mixing of information in the results section. I truly believe that the authors could benefit from taking a time to reorder the text, choosing more carefully where each piece of information will be placed, and maybe using more sub-sections to sort it out. They can take advantage of tables to summarize information too. Here, a question remains to the authors: how can you present the story behind the contradictions without making your reader feel lost? POINT 11) Lines 362-365: The authors are calling the groundwater in the basin "an uncertainty". I think they should consider wording here. . . the groundwater cannot be called an uncertainty, in my opinion. You may consider rephrasing it to something like "the epistemic uncertainty related to the groundwater physical processes in the basin". POINT 12) Lines 421-423: authors should provide scientific references here to support their argument that the GRACE data is not suitable in this case because of its resolution."

Response Point 10: See point 3A

Response Point 11: We consider this a valid point. We will change that in the revised section of the paper.
Response Point 12: We consider this a valuable observation. We have found the following reference stating the same problem of resolution, which we will include in the revised version of the paper: Castellazzi, P., Longuevergne, L., Martel, R., Rivera, A., Brouard, C., & Chaussard, E. (2018). Quantitative mapping of groundwater depletion at the water management scale using a combined GRACE/InSAR approach. Remote sensing of environment, 205, 408-418.

> Is the description of experiments and calculations sufficiently complete and precise to allow their reproduction by fellow scientists (traceability of results)?

"The authors declared having all required materials and model available online to run the reported simulations. In fact, their web link is working well, although I did not have the time to download it and try it by myself. However, as I commented previously, the methods section lacks details about model description, table of main parameters, source information about the dataset used to run the simulations, etc."

> Do the authors give proper credit to related work and clearly indicate their own new/original contribution?

"Yes."

> Does the title clearly reflect the contents of the paper?

"Yes. Totally."

> Does the abstract provide a concise and complete summary?

"Although the abstract is well written and summarizes very well the content of the paper, I would argue that it is too long for an abstract and the authors would benefit from writing everything in the paper (not only the abstract) more concisely and direct to the point. Tiny detail - Line 2: nobody "owns" a model in your case. A freeware was used to run the simulations. You could say, instead, that the modelling process was dominated by a particular group."

Response: We understand your point, and we think it is a matter of interpretation of the word own. We did not intend to use the literal meaning of "own", but the figurative meaning of a sense of ownership by the stakeholders. Anyway, we understand the confusion and will make it clearer in the revised version of the paper.

> Is the overall presentation well-structured and clear?

"In my opinion, although the paper presents a very interesting case and it is worth publishing, the major drawback here is, in fact, presentation and structure. The text is too dense, it repeats in some parts, information is not well distributed, nor a good use of structures (such as tables/figures/diagrams) was made to summarize and facilitate the reading. I strongly advise the authors to consider improving the structure and readability of the paper."

Response: We will change some sections and add a timeline.

> Is the language fluent and precise?

"The language is fluent and there are just a few typos. Every piece of information is well understood. However, again, the writing should be straight to the point in order to improve readability."

Response: We will revise the paper again to find the typos.

> Are mathematical formulae, symbols, abbreviations, and units correctly defined and used?

"Yes."

> Should any parts of the paper (text, formulae, figures, tables) be clarified, reduced, combined, or eliminated?

"In my comments above I emphasized the need to reduce, clarify, and reorder the text into more meaningful sections. Particularly in terms of reducing the text, I would advise the authors to apply this to the abstract and the conclusion sections."
Response: We will condense these sections in the revised version of the paper.

> Are the number and quality of references appropriate?

"Yes."

> Is the amount and quality of supplementary material appropriate?

"According to my comments about the methods section, additional information about the model, parameters and data sources could be provided in the form of a supplementary material."

Response: See our response for point 4.

We again want to thank Leticia for her valuable comments.

Best regards,

Jonatan, Nora and Pieter.

---

## Author Response (AR1)

**Authors´ general response: We want to thank the reviewers and the editor for their valuable comments and constructive criticisms, which gave us the opportunity to significantly improve the quality of the manuscript.**

The manuscript has been reviewed by two reviewers and myself, with responses to the reviewer comments submitted by the authors. Since the reviewers saw potential in the manuscript I will invite a revision, but the manuscript needs much additional work to fulfil its claims.

As a study on the entanglements of science and politics, the manuscript remains superficial. I suggest a much closer reading of Science and Technology Studies – a field that studies exactly what the authors wish to do – and Political Ecology. In particular, the following papers may serve as templates for how to write such a "story" (reviewed in Krueger et al., 2016): Alatout, 2013; 2014; Bouleau, 2014; Budds, 2009; Deroubaix, 2008; Fernandez, 2014; Forsyth, 2008; Mehta, 2010; Milman and Ray, 2011; Zimmerer, 2008.

As evidenced by these papers, the uncertainty frame the authors chose is sensible (despite some confusion discussed below), but the authors must be careful not to reduce the case to an epistemic problem. This was highlighted by Reviewer 2. He suggests analyzing the positions and work practices of those making choices in the production of hydrological knowledge in this politically charged situation. This is exactly what studies analyzing the entanglements of science and politics do (see for example Milman and Ray, 2011). In their response the authors note that they lack empirical material on these points. In this case the claims of the paper should be adjusted; as a study on the entanglements of science and politics it leaves too many questions unanswered. Analyzing the stakeholder reactions to the hydrological knowledge (model) produced (as also suggested by Reviewer 2) will go some way towards a reframing of the paper.

**Authors´ response: After reflecting on the editor´s and the second reviewer´s comment, we agree that we needed to improve throughout the manuscript how actors reflected on the controversies and the results of the UNOPS' study to provide more depth to the analysis; as well as to improve the structure of the story as suggested by the first reviewer.**

**We decided to improve the readability and the message of the manuscript with the following changes. First, we modified the structure of the text and added a general description of the study areas under Section 3, as suggested by the first reviewer. Second, we improved and moved the text that was formerly as 3.1 Case Study to the Results section "4.1 The Zapotillo conflict". Moreover, we improved this same section by adding a timeline in Figure 3 to facilitate the understanding of the case story. Moreover, after analyzing our field notes, we also expanded on the description of the back story behind the hiring UNOPS in L305-322, as well as in the description of the two controversies  Third, to supplement the absence of interviews with the UNOPS team, we thoroughly checked again their scientific report to provide a quote on the reasonings behind their modeling decisions (L480-488). Finally, we added the impressions of the actors after hearing the results and recommendations of the UNOPS team (L575-593).**

**Regarding the STS studies the editor recommended, we had already reviewed the paper by Budds (2009) in our paper. However, we thoroughly read the other recommended papers. Below we summarize the key elements of the papers that we mobilized to improve readability of the paper, as well as support our analytical approach and/or findings.**

Fernandez (2014) describes the history of water management in the Garonne watershed in France to highlight how the use of indicators erases the context-specific circumstances in which they were created. By using a Foucauldian approach, the author unveils interests that molded scientific practices, which were naturalized in a way that hid the adversarial points of view of different actors. The study shares some analytical elements with ours, such as the use of controversies, claims of knowledge and the use of black-boxing models that hide interests.

Milman & Ray (2011) discuss how uncertainties related to groundwater can produce ambiguities to support practices and interests at each side of the US-Mexico border. The paper analyses transboundary groundwater management between Mexico and USA. For Mexico, the objective is to provide water for human use and improve livelihoods regardless of environmental uses; while, for USA, the objective is to balance the aquifer also for environmental considerations. They described how both countries differ in their appreciation of how the groundwater system works, driven by the inherent uncertainties of the system. They conclude how these appreciations are explained by the values of each country. We improved our manuscript by using a similar structure regarding the water availability in Los Altos, and how two different interests interpret this epistemic uncertainty, as well as reproducing verbatim comments by the actors.

Bouleau (2014) describes how the understanding of different water systems in France and their problems were the result of context-specific scientific developments that prioritized certain solutions and marginalized others. Consequently, two hydrological regions in France developed different ways to understand the water system.

Forsyth (2011) examines the possibility of acknowledging social influence on scientific explanations and also developing situated knowledge that can explain causal relationships between environmental systems and society. To examine that possibility, he analyzed soil erosion in Thailand, and the way that the universal soil erosion loss equation played a role in wrongly identifying local agricultural practices as the culprits of erosion. Forsyth found that local practices actually avoided soil erosion and that there were nonagricultural causes of erosion. These findings contributed to the potential of political ecology to provide better causal explanations of environmental problems.

Zimmerer (2008) offers a counter-explanation of the Cochambamba war rather than a simplistic explanation as caused by scarcity. The author analyses the standardization of spatial-geographic frameworks to understand scarcity in a basin and decide on irrigation projects. These analysis reinforce and even construct the portrayal of regions of abundance and scarcity. The author recommends the use of social-environmental models, as well as participatory social processes and planning.

These papers arrive at conclusions that sometime converge and sometimes differ, depending on the case at hand; which we try to summarise here. Forsyth (2011) concludes that knowledge needs to be situated to have more impact, and that if political ecology intends to remain relevant, it needs to go beyond how knowledge is generated and legitimized, and be able to also offer explanatory considerations of environmental issues. Bouleau (2014) concludes that the hydrosocial cycle is a useful concept to understand the dialogical relationship between the generation of science and the creation of waterscapes. Budds (2009) concludes that scientific assessments are never neutral and always uncertain to some degree; taking this into account

she developed the hypothesis that considering local knowledge  in the groundwater assessment of La Ligua, Chile by water authorities would have produced an alternative solution even though it may not have  changed the outcome. Zimmerer (2008) reaches a similar conclusion in the case of Bolivia. We find these conclusions similar to the conclusion of our manuscript, in which a more collaborative scientific production would have opened also alternative solutions.

Our take aways from the above references are some story cues (such as graphic timelines) and similarities in structure to improve the readability of the paper. From Fernández (2014), and also from the comment by reviewer 1, we incorporated a graphic timeline to our manuscript. Specifically, on the structure followed by most of the papers, from Budds, (2009), Milman & Ray (2011), Fernandez (2014) we have adopted a brief introduction of the study areas into our manuscript; an improvement also suggested by reviewer 1. Then, we also present a critical history of the water developments in the study areas, which brings socio-political context to the environmental problem being analyzed. Thereafter, we follow the same pattern as most papers suggested by the editor; since we critically analyze the adversarial knowledge claims by different and opposing actors, the inherent uncertainties in the water system, and how hegemonic actors instrumentalize alleged scientific knowledge to close the debate and impose/advocate certain solutions to the environmental problem.

In discussing their empirical material, the authors recommend a participatory approach to knowledge production and water management. As noted by both reviewers, this recommendation lacks awareness of the shortcomings of participatory processes, especially in development contexts (e.g. Blaikie, 2006; Cooke and Kothari, 2001). The problem is that the authors don't have any empirical material on participation, they can only diagnose a lack of participation in their case and speculate about what this means for the dilemma at hand and how it might be resolved though more participation. This does not contribute sufficiently to the literature. What I would suggest is that the authors foreground the stakeholder perspective as part of the science/policy entanglement (as advised above based on Reviewer 2) and stay close to the empirical material rather than speculating about the success of hypothetical participatory processes.

Authors´ response: We reviewed the texts provided by the reviewer and concluded the following. First, Blaikie (2006) discusses decentralization and public participation, but in the context of community-based natural resources management in Africa. We think the cases discussed by Blaikie are very different from the Zapotillo case. Before the Zapotillo project was announced there was no conflict over the water resources in the Verde River Basin; so the conflict is not on how to manage locally the water from the basin, but on imposing a large infrastructure project. Therefore, we think is not possible to make a valid comparison between the Zapotillo and the African cases.

From the book The Tyranny of Participation (Cooke & Kothari, 2001), we analyzed the chapter by Mosse "´People´s Knowledge', Participation and Patronage: Operations and Representations in Rural Development." We also found that this chapter cannot be directly compared with the Zapotillo case. Mosse revised rural development cases in India, and although he has a compelling story on how local knowledge reflects the interests of the funding agencies, dominant groups and project donors, these are clearly cases of participation in contexts of development cooperation, not contexts of conflict. According to Mosse´s reflection of the case, local actors manipulate planning knowledge to advance their own interests in a way that can seem legitimate by the project managers; while project planners, involved in the logic of

**delivering objectives overlook this and unintentionally reinforce patronage networks and hierarchical modes of operation. However, in the case of the Zapotillo case, the local actors´ positions and their own interests are public and do not intend to legitimize their positions to any donor agency; it is hard to imagine that people from Temacapulín can be influenced by powerful groups (water authority, construction companies, etc.), since their main interest is to save their town from being flooded.**

**We understand that it is better to give cautious recommendations, especially when it is not backed by empirical material. However, it is worth noting that the alternative approach (a lack of participation) has already failed. What we can conclude, in a similar way as Zimmerer (2008) and Budds (2009) concluding hypothesis, is that the inclusion of participation in the production of knowledge, although it would probably not have changed the decision, it would have at least opened alternative solutions.**

**In order to be more cautious in our recommendations we rephrased to clarify this as a hypothesis. We replaced the original text for this one (L745-762): "Returning to the original question whether science can depoliticize conflicts or ifwhether science is politicized in the process, this case has shown that attempting to depoliticize science-policy processes is very difficult, since these processes are inherently political. Moreover, involving alleged neutral - or apolitical - third parties to depoliticize scientific knowledge to resolve water conflicts can backfire if they act - or are perceived - as stealth advocates of political interests. However, we identified two elements that can contribute to a possible transformation of the conflict and management of such politicization. First, scientist in contexts of conflict should be aware of not promoting specific solutions, since that is the role of the political actors, When scientists assume the role of "honest broker of policy alternatives" (Pielke, 2007), it restrains them from offering a specific course of action and compels them to expand the scope of choice for the actors in the conflict. And second, to promote social mechanisms to filter as much as possible which knowledge claims are more value-laden, and which are less, in particular in contexts of conflict and high uncertainties. There is an urgent need to design water resources models in a more open way to allow the participation of stakeholders and legitimize the data used in them (Islam & Susskind, 2018) as well as the values hidden in them ; t. This can support the necessary task of reviewing alternatives to large infrastructures (Van der Zaag & Gupta, 2008). Additionally, fostering stakeholder participation could collaboratively bring about socially relevant research questions that open the decision space (Voinov & Gaddis, 2008; Zimmerer, 2008; Budds, 2009; Lejano & Ingram, 2009; Brugnach et al., 2011; Blöschl et al., 2013; Armitage et al., 2015; Basco-Carrera et al., 2017; Van Cauwenbergh et al., 2018; van der Molen, 2018; Norstöm et al., 2020). However, since participation could present some pitfalls (i.e. Mosse, 2001; Godinez Madrigal et al., 2019), Krueger et al. (2016) recommend to test each actor´s claims and preconceptions through object-based processes (i.e. maps and models, see also Brugnach & Ingram, 2012) to co-produce knowledge beyond discourse."**

Reviewer 1 made further helpful suggestions for improving the structure of the paper. As part of this the authors should include more details on the interviews conducted and the subsequent analysis of the empirical material (as also suggested by Reviewer 1). Including the interview guides for the semi- structured interviews in the Appendix and information on coding would seem especially important.

**Authors´ response: We added additional text in the method section to explain better how we conducted the interviews: "Due to the delicate nature of the situation, all interviewees remain anonymous, and not all interviews could be recorded; in such cases we relied on fieldnotes taken immediately after the interview. The interviews that were recorded, were transcribed. We analyzed the interview transcripts and fieldnotes to extract the summarized viewpoint of the stakeholders, which are described in Table 1."**

**About the interview guides, we mentioned in the manuscript that "the semi-structured interviews consisted of exploring three main themes: the root causes of the problem and the conflict, what were the sources of controversy in the conflict, and what would be the preferred solutions to the conflict and the water scarcity problem." Since sub-sequent questions were tailored to each interviewee, revealing those questions would compromise the anonymity of them. So, in compliance with our research ethics, we would prefer not to include the guides nor the coding we use to analyze them.**

In addition, I had the following comments:

L71-77: The uncertainty frame is helpful but only as far as the role of uncertainty in science/policy relations is concerned, not as the root cause of the problem in the present case (compare Reviewer 2). There is also a misunderstanding of aleatory and epistemic uncertainty; aleatory is the one conceived of as irreducible. Scientific uncertainty does matter as it allows the same piece of evidence to be interpreted differently for different political ends (e.g. Milman and Ray, 2011). But the real challenge seems to be value disagreement (to speak with Funtowicz and Ravetz). It would seem more fruitful to analyze knowledge claims and ask how they are produced, what they leave out, what authority they enjoy and why and how they have political consequences.

**Authors´ response: We consulted again the work by Di Baldassarre et al. (2016), and indeed we found that aleatory uncertainty is considered irreducible, albeit manageable with probabilistic methods: " While the exact time of occurrence of future flood events cannot be deterministically predicted, this intrinsic uncertainty can be assumed to be predominantly aleatory and can easily be treated in probabilistic terms."**

**Concerning the value disagreement issue, we reviewed our fieldwork material, and found that in our interviews, some actors agreed on the difference of values, which contributes to the political side of the conflict. So, we added the following text under the section 4.2.2 Negative consequences for the donor basin (L490-493): "Regarding the dam´s height and the three communities under threat of displacement, the controversy lies in incompatible values that have different legal and technical consequences. The Temacapulín community reasserted its human rights of consult and consent, participation, and cultural and historical heritage; while the government of Jalisco reasserted the utilitarian argument of the greatest good to the largest number of people over the right of a few ones."**

**On the last comment about analyzing knowledge claims, we actually did that. In section 4.3 we analyzed the knowledge claims by IMTA, Conagua, Observatory and the animal farmer´s association. We precisely analyzed what they left out from each study and its political consequences.**

**Authors´ response: We changed the wording of the sentence to replace non-expert knowledge to "stakeholders without an academic background to be included in research and decision making"**

Fig3: The differences between the scenarios should be explained better.

**Authors´ response: We improved the description of the five scenarios in L199-205.**

Section 4: More should be made of the authors' own modelling study. Why was the scenario they created omitted in the official study? With what consequences?

**Authors´ response: As we mentioned previously, we added a quote of UNOPS´ study, where they justify the choice of scenarios: "Although UNOPS' team could have developed many other scenarios with different variables, the report of the study justified choosing only these five scenarios in the following way "the definition of the number of scenarios is not absolute, but may be subject to future changes at any time that it is required to attend to different questions from those raised in the framework of this study […] Specifically, it is interesting to know under which configuration of the dam´s height and volume of water transfer can guarantee water demand and what percentage of satisfaction corresponds to it, which leads to justifying technically the presence of the dam and its geometric configuration (UNOPS, 2017b: 27-28)."**

**Then, we finalized the section with the following explanation: "Therefore, the poor results of these indicators do not seem to justify the implementation of the Zapotillo project as it is currently designed."**

**Moreover,  in the discussion section we explore the consequences of omitting this scenario: "According to the best social and hydrological knowledge available, it can be inferred from our scenario that there are insufficient surface water resources to satisfy the demand of the three regions' explosive demographic and economic growth, which means that at least one region will continue to unsustainably deplete its groundwater resources. In fact, UNOPS fifth scenario generated positive results only because it considered null demographic and economic growth for the future and did not consider climate change in the Verde River Basin."**

L409-410: Epistemic uncertainties are (partly) about accuracy and precision.

**Authors´ response: Yes, we agree. We added to the text ", which partially addressed epistemic uncertainties"**

L426: It is not readily evident form the empirical material that this is a case of an epistemic controversy. A general framing of science/politics entanglements will be better suited.

**Authors´ response: We argue that it is both, because although the entanglement of science and politics is clearly present in the case, there are latent/emerging problems that  have not been addressed partly due to epistemic uncertainties, and the overwhelming focus given to the Zapotillo project. For example, it is still unclear what the real extent of groundwater overexploitation in the Verde River Basin is. In order to make that clearer, we added in the conclusions the following text: "This study has two main findings. 1) Intractable water conflicts tend to isolate the process of knowledge production, which foregrounds issues that are politically convenient for each actor, while other issues, perhaps more important for sustainability (like groundwater over-exploitation), remain unaddressed."**

L434: Participation is not only about impact, but also about substance and ethics.

**Authors´ response: We agree to this comment. We changed the text in the following way: "not only its impact, but also to better policies and contravene the rights of stakeholders to participate (Krueger et al., 2016)."**

L434: Note the large body of literature on participation in a development context (e.g. Blaikie, 2006; Cooke and Kothari, 2001).

**Authors´ response: We have answered this issue above in Page 3.**

Section 6: Many of the claims made are not substantiated by empirical material; the authors should be careful to stick with the case material and not speculate beyond it (compare Reviewer 1).

**Authors´ response: We re-considered some of our claims, especially in the third paragraph of section 6.**

L469: How exactly science and politics are entangled does not become clear from the case study. L470-471: That the case is one of knowledge controversies remains equally unclear.

**Authors´ response: We added the impressions of the actors to the UNOPS study and analyzed them through the work of Pielke, in which the UNOPS' team was perceived to have acted as stealth issue advocates. We think that this addition clarifies how science and politics were entangled in the Zapotillo case. These responses make it clear how the key stakeholders perceived the study of UNOPS as a political maneuver rather than a "pure" scientific product.**

**For the second comment, we respectfully disagree for reasons we have explained under the editor´s comment of L426.**

L473-475: Here the authors foreground the epistemic problem, which is only part of the story (compare Reviewer 2).

**Authors´ response: We agree with this comment. We modified the text as follows:**

**"The conflict is defined by epistemic uncertainties, ambiguity, and incompatibility of values. The first two consist of several knowledge controversies regarding water availability and the negative effects of the water transfer and dam construction in the donor basin, and the possible alternatives to supply augmentation strategies in the recipient basins. The latter consists of a dispute over the distribution of the environmental, social and economic costs and benefits derived of the Zapotillo project."**

L493-494: This belief in the power of science and participation is unjustified, see basic texts like Pielke (2007).

**Authors´ response: We thoroughly read Pielke´s text. The book states that science alone cannot provide concrete courses of action, since these are almost exclusive of politics and policy, otherwise there is a risk of 'technocracy' or 'scientization' (p.35). However, the author also provides some promising reflections on the relationship between public participation and**

science. On page 114, Pielke states that in contexts of deep uncertainty and conflict, public policies derived from public participation make more sense than large-scale commitments.

In the Zapotillo case, we described how actors engaged in the "manufacture of ignorance", which is also described by Pielke (2008: 63) as inventing "facts" as convenient. Since these questionable facts enhanced the intractability of the conflict, we think it is justified to assert that scientific information has a role to play in the policy process of the Zapotillo conflict, but only as a function of democratic and participative decision-making as stated by Pielke (2007: 37). Also Krueger et al. (2016) recommends to put knowledge (of all actors participating in knowledge production) to the test to keep up with the scientific spirit.

However, we agree that our text may not have been clear on this, and may have sounded overly optimistic about the power of science. Therefore, we rephrased the lines in the text: "Returning to the original question whether science can depoliticize conflicts or whether science is politicized in the process, this case has shown that attempting to depoliticize science-policy processes is very difficult, since these processes are inherently political. Moreover, involving alleged neutral - or apolitical - third parties to depoliticize scientific knowledge to resolve water conflicts can backfire if they act - or are perceived - as stealth advocates of political interests. However, we identified two elements that can contribute to a possible transformation of the conflict and management of such politicization. First, scientist in contexts of conflict should be aware of not promoting specific solutions, since that is the role of the political actors, When scientists assume the role of "honest broker of policy alternatives" (Pielke, 2007), it restrains them from offering a specific course of action and compels them to expand the scope of choice for the actors in the conflict. And second, to promote social mechanisms to filter as much as possible which knowledge claims are more value-laden, and which are less, in particular in contexts of conflict and high uncertainties. There is an urgent need to design water resources models in a more open way to allow the participation of stakeholders and legitimize the data used in them (Islam & Susskind, 2018) as well as the values hidden in them ; t. This can support the necessary task of reviewing alternatives to large infrastructures (Van der Zaag & Gupta, 2008). Additionally, fostering stakeholder participation could collaboratively bring about socially relevant research questions that open the decision space (Voinov & Gaddis, 2008; Zimmerer, 2008; Budds, 2009; Lejano & Ingram, 2009; Brugnach et al., 2011; Blöschl et al., 2013; Armitage et al., 2015; Basco-Carrera et al., 2017; Van Cauwenbergh et al., 2018; van der Molen, 2018; Norstöm et al., 2020). However, since participation could present some pitfalls (i.e. Mosse, 2001; Godinez Madrigal et al., 2019), Krueger et al. (2016) recommend to test each actor´s claims and preconceptions through object-based processes (i.e. maps and models, see also Brugnach & Ingram, 2012) to co-produce knowledge beyond discourse."

[revised manuscript text omitted]

Acción Colectiva | Guanajuato's government and State water authorities

Municipality of León and SAPAL

Guanajuato's business association |
| **Jalisco** | ITESO

Universidad de Guadalajara

Temacapulin

IMDEC

Colectivo COA

Animal farmers' association of Los Altos

Observatory

Municipality of Cañadas de Obregón | Universidad Panamericana

Universidad Autónoma de Guadalajara

Chamber of the industry of construction of Jalisco

College of Civil Engineers of Jalisco

SIAPA

Jalisco's government and State water authorities |
| **Federal** | | Conagua

IMTA |

[revised manuscript text omitted]

1. This data is from 2015, the latest official data available.
2. The overexploitation counts all aquifer users, including agriculture and industry.
3. We consider only the municipalities of Guadalajara, Zapopan, Tonalá and Tlaquepaque.

**Supplementary material 2. Detailed information on the key actors in the conflict.**

Figure 4 show the position of key actors vis-à-vis the Zapotillo dam project. Each actor's position depended on some factors described in Table S2. If an actor is a key and direct stakeholder of the project, lobbied for or against the project, and has publicly condemned or be in favor of the project, it is at the extreme of the spectrum.

**Table S2. Detailed information on the key actors in the conflict.**

[revised manuscript text omitted]

**Supplementary material 3. Detailed information of the UNOPS´ Verde River Basin model.**

UNOPS model of the Verde River basin is developed in WEAP software. This is a water planning software, functioning with the principle of water balance accounting. The software analyses the diverse water supply sources, as well as the withdrawal and transmission to water demand nodes. To start the analysis the software needs a "time frame, spatial boundary, system components and the configuration of the problem." (WEAP, 2020). The software uses two main features to analyze the water resources system. One, called 'Current accounts' analyses the present water demand, resources and supplies based on economic, demographic, hydrological trends to present a snapshot of a business-as-usual scenario. The other explores scenarios to evaluate different strategies such as supply augmentation or demand management.

To create the model, it is necessary to delimit the area and establish the system boundaries. UNOPS first delimited the study area to that of the Verde River Basin, which was 21,495 km$^2$. The main natural variables that condition the models is percolation, precipitation, run-off, evapotranspiration, infiltration and interflow, while the variables derived from manmade interventions are reservoirs, groundwater draft, water transfers, water demand, derived flows and return flows. UNOPS used the data of Table S3 to fill these variables. The basic parameters used by

WEAP are the monthly variation of demand, climate data, and then uses the MABIA water balance method to compute the water balance. This method is based on the two-bucket structure that processes the root zone as the top bucket and what is below the root zone as the bottom bucket (representing groundwater). The model proceeds to process 8 necessary steps to compute the water balance: 1) reference evapotranspiration, 2) soil water capacity, 3) basal crop coefficient, 4) evaporation coefficient, 5) potential and actual crop evapotranspiration, 6) water balance of the root zone, 7) irrigation, and 8) yield. Groundwater is calculated through nodes that compute the natural recharge flows (the top bucket), demand returns, infiltration losses from aqueducts and reservoirs and river recharges as flows that replenish the groundwater storage, and groundwater draft and base flows are computed as flows that deplete groundwater storage.

Then, since the software is configured to create semi-distributed models, UNOPS created 18 sub-regions that were characterized by a similar climate by using data mentioned in Table S3 through GIS extrapolation procedures. To ascertain the validity of all this sub-regions, UNOPS integrated the following data layers of all 18 sub-regions: 1) the hydrographic network, 2) the artificial regulation and monitoring system (dams, sluices and hydrometric stations), 3) the overlaps of the limits of aquifers with the density of surface and groundwater extractions, 4) soil and land cover characteristics, and the climatic distribution within the basin (UNOPS, 2017).

For the first layer, UNOPS used specialized algorithms to process in a digital elevation model a hydrographic network. For the second layer, the model used CONAGUA´s observed data of 7 reservoirs (flows, volume and storage levels) and 8 hydrometric stations; this data was used to calibrate the model. For the third layers, CONAGUA´s georeferenced database of water rights (REPDA) was used to determine hotspot areas of groundwater draft. With the fourth layer the model was able to process the relation with the upper and bottom buckets of the MABIA water balance method, and used SGM´s, SSN´s, INEGI´s and CONAFOR´s data for the soil characteristics, and CLCICOM´s and INIFAP´s data for the climatic distribution.

**Table S3. UNOPS´ Verde River Basin model variables (source: UNOPS, 2017).**

| | Source | Spatial resolution | Temporal resolution |
|---|---|---|---|
| **Climate** | CLCICOM (CONAGUA-SMN) | 315 stations including a buffer of 50 km outside the contour of the basin | Daily (1943-2014) |
| **Hydroclimatology** | INIFAP | 105 stations including a buffer of 50 km outside the contour of the basin | Monthly (2002-2014) |
| **Hydrometry in rivers and reservoirs** | BANDAS (CONAGUA) | Timeseries of 8 hydrometric stations | 1941-2016 |
| **Groundwater** | CONAGUA | Studies of 21 aquifers of the region. | 1997-2010 |
| | GRACE (GFZ German Research Centre for Geosciences, Center for Space Research - The University of Texas at Austin and NASA Jet Propulsion Laboratory) | Monthly fields with gravity coefficients at a spatial resolution of 1ºx1º ($\approx$ 11,000 km$^2$) | 2002-2014 |
| **Soil and Land cover** | INEGI, CONAFOR | Land cover maps 1:50,000, irrigation district maps 1:250,000, and images of SPOT 6 and 7 | 2012-2016 |
| **Water demand** | REPDA (CONAGUA) | All georeferenced surface and groundwater rights | 2016 |
| **Geology** | SGM, SSN | Maps 1:50,000 and 1:250,000 | 2007 |

| | | | |
|---|---|---|---|
| **Population/returns** | INEGI | Population of towns with more than 2,500 people | 2010 |
| **Digital Elevation Model** | CEM v2.0 | Raster with resolution of 15m (1:20000) | 2013 |

---

## Editor Decision (ED1)

The manuscript has been reviewed by two reviewers and myself, with responses to the reviewer comments submitted by the authors. Since the reviewers saw potential in the manuscript I will invite a revision, but the manuscript needs much additional work to fulfil its claims.

As a study on the entanglements of science and politics, the manuscript remains superficial. I suggest a much closer reading of Science and Technology Studies – a field that studies exactly what the authors wish to do – and Political Ecology. In particular, the following papers may serve as templates for how to write such a "story" (reviewed in Krueger et al., 2016): Alatout, 2013; 2014; Bouleau, 2014; Budds, 2009; Deroubaix, 2008; Fernandez, 2014; Forsyth, 2008; Mehta, 2010; Milman and Ray, 2011; Zimmerer, 2008.

As evidenced by these papers, the uncertainty frame the authors chose is sensible (despite some confusion discussed below), but the authors must be careful not to reduce the case to an epistemic problem. This was highlighted by Reviewer 2. He suggests analyzing the positions and work practices of those making choices in the production of hydrological knowledge in this politically charged situation. This is exactly what studies analyzing the entanglements of science and politics do (see for example Milman and Ray, 2011). In their response the authors note that they lack empirical material on these points. In this case the claims of the paper should be adjusted; as a study on the entanglements of science and politics it leaves too many questions unanswered. Analyzing the stakeholder reactions to the hydrological knowledge (model) produced (as also suggested by Reviewer 2) will go some way towards a reframing of the paper.

In discussing their empirical material, the authors recommend a participatory approach to knowledge production and water management. As noted by both reviewers, this recommendation lacks awareness of the shortcomings of participatory processes, especially in development contexts (e.g. Blaikie, 2006; Cooke and Kothari, 2001). The problem is that the authors don't have any empirical material on participation, they can only diagnose a lack of participation in their case and speculate about what this means for the dilemma at hand and how it might be resolved though more participation. This does not contribute sufficiently to the literature. What I would suggest is that the authors foreground the stakeholder perspective as part of the science/policy entanglement (as advised above based on Reviewer 2) and stay close to the empirical material rather than speculating about the success of hypothetical participatory processes.

Reviewer 1 made further helpful suggestions for improving the structure of the paper. As part of this the authors should include more details on the interviews conducted and the subsequent analysis of the empirical material (as also suggested by Reviewer 1). Including the interview guides for the semi-structured interviews in the Appendix and information on coding would seem especially important.

In addition, I had the following comments:

L71-77: The uncertainty frame is helpful but only as far as the role of uncertainty in science/policy relations is concerned, not as the root cause of the problem in the present case (compare Reviewer 2). There is also a misunderstanding of aleatory and epistemic uncertainty; aleatory is the one conceived of as irreducible. Scientific uncertainty does matter as it allows the same piece of evidence to be interpreted differently for different political ends (e.g. Milman and Ray, 2011). But the real challenge seems to be value disagreement (to speak with Funtowicz and Ravetz). It would seem more fruitful to analyze knowledge claims and ask how they are produced, what they leave out, what authority they enjoy and why and how they have political consequences.

L82: It would be naïve to think there could ever be a fair assessment of different kinds of knowledge (compare both reviewers).

L86: But why exactly does science have this authority and how exactly is it entangled with power?

L88: There is a lot more to say about bottom-up or participatory or transdisciplinary approaches; they are not just aiming at reducing epistemic uncertainty, and even if they did there is enough critical literature on the limits of achieving this aim.

L121: This is a limited reading of Krueger et al., 2016. The paper is not advocating non-expert knowledge per se, but argues for people who are not scientists to get involved with science for epistemic, political and ethical reasons. It bases this argument on a review of case studies of the entanglements of hydrological science and politics.

Fig3: The differences between the scenarios should be explained better.

Section 4: More should be made of the authors' own modelling study. Why was the scenario they created omitted in the official study? With what consequences?

L409-410: Epistemic uncertainties are (partly) about accuracy and precision.

L426: It is not readily evident form the empirical material that this is a case of an epistemic controversy. A general framing of science/politics entanglements will be better suited.

L434: Participation is not only about impact, but also about substance and ethics.

L434: Note the large body of literature on participation in a development context (e.g. Blaikie, 2006; Cooke and Kothari, 2001).

Section 6: Many of the claims made are not substantiated by empirical material; the authors should be careful to stick with the case material and not speculate beyond it (compare Reviewer 1).

L469: How exactly science and politics are entangled does not become clear from the case study.

L470-471: That the case is one of knowledge controversies remains equally unclear.

L473-475: Here the authors foreground the epistemic problem, which is only part of the story (compare Reviewer 2).

L493-494: This belief in the power of science and participation is unjustified, see basic texts like Pielke (2007).

References

Alatout S. Revisiting water politics and policy in Israel: policymaking under conditions of uncertainty. In: Megdal SB, Varady RG, Eden S, eds. Shared Borders, Shared Waters: Israeli–Palestinian and Colorado River Basin Water Challenges. Boca Raton, FL: CRC Press; 2013, 75–89.

Alatout S. From river to border: the Jordan between empire and nation-state. In: Kleinman DL, Moore K, eds. Routledge Handbook of Science, Technology, and Society. Oxon: Routledge; 2014, 307–331.

Bouleau G. The co-production of science and waterscapes: the case of the Seine and the Rhône Rivers, France. Geoforum 2014, 57:248–257.

Budds J. Contested H2O: science, policy and politics in water resources management in Chile. Geoforum 2009, 40:418–430.

Cooke B, Kothari U, eds. Participation: The New Tyranny? London: Zed Books; 2001, 207.

Blaikie P. Is small really beautiful? Community-based natural resource management in Malawi and Botswana. World Dev 2006, 34:1942–1957.

Deroubaix JF. The co-production of a "relevant" expertise—administrative and scientific cooperation in the French water policies elaboration and implementation since the 1990s. Hydrol Earth Syst Sci 2008, 12:1165–1174.

Fernandez S. Much ado about minimum flows… unpacking indicators to reveal water politics. Geoforum 2014, 57:258–271.

Forsyth T. Politicizing environmental explanations: what can political ecology learn from sociology and philosophy of science?. In: Goldman MJ, Nadasdy P, Turner MD, eds. Knowing Nature: Conversations at the Intersection of Political Ecology and Science Studies. Chicago, IL: The University of Chicago Press; 2008, 31–46.

Krueger, T., Maynard, C., Carr, G., Bruns, A., Mueller, E.N. and Lane, S. (2016), A transdisciplinary account of water research. WIREs Water, 3: 369-389. doi:10.1002/wat2.1132

Mehta L. The Limits to Scarcity: Contesting the Politics of Allocation. Earthscan: Oxon; 2010.

Milman A, Ray I. Interpreting the unknown: uncertainty and the management of transboundary groundwater. Water Int 2011, 36:631–645.

Pielke R. 2007. The Honest Broker: Making Sense of Science in Policy and Politics. Cambridge University Press.

Zimmerer KS. Spatial-geographic models of water scarcity and supply in irrigation engineering and management: Bolivia, 1952–2009. In: Goldman MJ, Nadasdy P, Turner MD, eds. Knowing Nature: Conversations at the Intersection of Political Ecology and Science Studies. Chicago, IL: The University of Chicago Press; 2008, 167–185.

---

## Author Response (AR2)

[revised manuscript text omitted]

1 (HD & CWD) 2 (HD & CWD) 3 (HD & CWD) 4 (CC & FWD) 5 (HD & CWD) Our (CC & FWD) Corresponding water allocations with different heights of the Zapotillo dam, precipitation and temperature patterns, and water demand

■ León ■ Guadalajara ■ Los Altos

105 m

105 m

105 m

4. Results

0

105 m

105 m

**4.1 The Zapotillo conflict**

- 220 Guadalajara and León are the most important cities of their respective States, Jalisco, and Guanajuato, in terms of population and economic size. Since the 1950s, Guadalajara's local water resources availability was overrun by the increasing water demand, and water managers sought to increase its water supply from Lake Chapala, the largest lake in the country. Currently, Guadalajara complements its water demand mainly through groundwater (see Table S1 in the supplementary material). However, due to their intense use, the aquifers are considered as over-exploited and with presence of nitrate and sulfatesulphate
- due to farming activities and wastewater disposal, and naturally occurring contaminants like lithium, manganese, fluorine, and barium due to mixing of hydrothermal fluids (Hernandez-Antonio et al., 2015; Mahlknecht et al., 2017; Moran-Ramirez., 2016). León, on the other hand, does not have large bodies of surface water in close vicinity and therefore it has historically relied solely on groundwater, which is now considered as heavily over-exploited with a drawdown of 1.5 m/year and with presence of chromium due to industrial activities, related to anthropogenic activities nitrate, chloride, sulfatesulphate, vanadium and pathogens, and naturally occurring contaminants like fluoride, arsenic, iron, and manganese due to the introduction of older groundwater with longer residence times (Esteller et al., 2012; Villalobos-Aragon et al., 2012; Cortes et al., 2015; SAPAL, 2020).